# 🤖 FACT-R1: Towards Explainable Video Misinformation Detection with Deep Reasoning

**Fanrui Zhang**[1,2*], **Dian Li**[3†], **Qiang Zhang**[1*], **Jun Chen**[3], **Gang Liu**[3],
**Junxiong Lin**[4], **Jiahong Yan**[3], **Jiawei Liu**[1*], **Zheng-Jun Zha**[1†]

[1]MoE Key Laboratory of Brain-inspired Intelligent Perception and Cognition, USTC
[2]Shanghai Innovation Institute    [3]Tencent QQ    [4]Fudan University
{zfr888, zq_126}@mail.ustc.edu.cn  {jwliu6, zhazj}@ustc.edu.cn
{goodli, albertjchen, sinbadliu, redyan}@tencent.com  linjx23@m.fudan.edu.cn

## Abstract

The rapid spread of multimodal misinformation on social media has raised growing concerns, while research on video misinformation detection remains limited due to the lack of large-scale, diverse datasets. Existing methods often overfit to rigid templates and lack deep reasoning over deceptive content. To address these challenges, we introduce FakeVV, a large-scale benchmark comprising over 100,000 video-text pairs with fine-grained, interpretable annotations. In addition, we further propose Fact-R1, a novel framework that integrates deep reasoning with collaborative rule-based reinforcement learning. Fact-R1 is trained through a three-stage process: (1) misinformation long-Chain-of-Thought (CoT) instruction tuning, (2) preference alignment via Direct Preference Optimization (DPO), and (3) Group Relative Policy Optimization (GRPO) using a novel verifiable reward function. This enables Fact-R1 to exhibit emergent reasoning behaviors comparable to those observed in advanced text-based reinforcement learning systems, but in the more complex multimodal misinformation setting. Our work establishes a new paradigm for misinformation detection, bridging large-scale video understanding, reasoning-guided alignment, and interpretable verification.

## 1 Introduction

Platforms such as TikTok and YouTube have significantly reshaped the landscape of news consumption by enabling the rapid and large-scale dissemination of short-form videos. While this shift facilitates fast and wide-reaching information exchange, it also dramatically accelerates the spread of misinformation, thereby eroding the credibility and integrity of public discourse. As manual fact-checking struggles to keep pace with the overwhelming volume and velocity of content, automated approaches have become increasingly essential for timely identification and mitigation of misinformation on social platforms [5, 24, 45, 47].

While recent advances in multimodal misinformation detection have focused primarily on image-text modalities [47, 7], the growing dominance of video content introduces new challenges. Video-text misinformation is not only more engaging and fast-spreading but also harder to detect due to the complexity of heterogeneous and temporally dynamic modalities [9, 48, 51]. The inherent complexity of video collection and annotation introduces several limitations in existing datasets [5]: (1) Limited scale and topic diversity, typically with only a few thousand videos collected over narrow timeframes

---

*  Equal contribution. † Corresponding author.

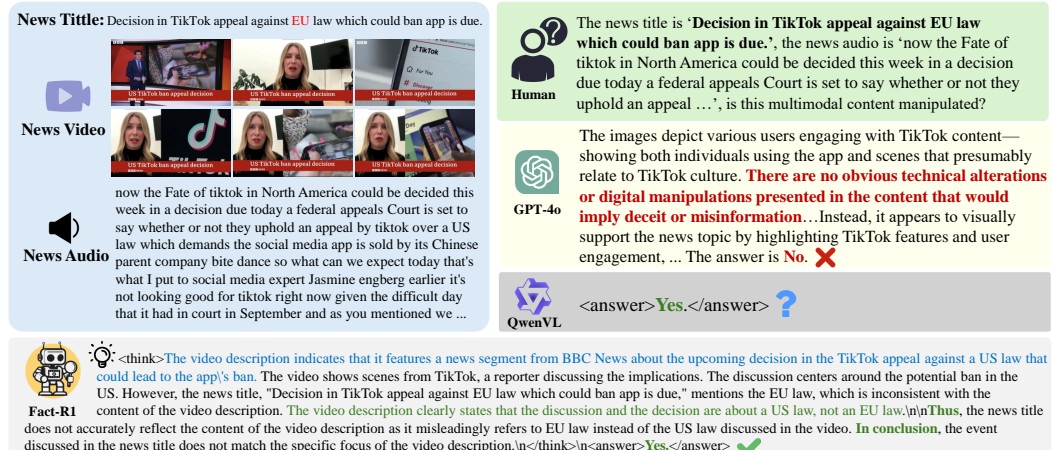

Figure 1: While state-of-the-art multi-modal models like GPT-4o fail to consistently detect video misinformation, and template-finetuned systems such as QwenVL remain constrained by rigid response formats, Fact-R1 establishes a novel paradigm by enabling deep, structured reasoning tailored for misinformation detection.

[21, 30]; (2) Lack of fine-grained annotations and standardized interpretability metrics, resulting in fragmented research and inconsistent evaluation [43, 6]; (3) The presence of fact-checking watermarks or logos in some fake videos, which risks models overfitting to superficial visual patterns instead of learning semantically meaningful cross-modal inconsistencies [29, 52, 18, 19]. Among the various forms of misinformation, one of the most prevalent and deceptive strategies is the Out-of-Context misuse of real videos—misleading audiences by presenting authentic footage within a fabricated or misleading narrative context [20, 46]. Owing to its high perceived credibility and low production cost, OOC has become a dominant tactic in the construction of persuasive misinformation campaigns [23, 33].

The emergence of Multimodal Large Language Models (MLLMs) has advanced the integration of visual and textual modalities, thereby steering the field of misinformation detection toward MLLM-based frameworks [25]. However, as illustrated in Figure 1, existing approaches still face critical limitations: (1) Many rely on rigid fine-tuning templates, resulting in overfitting to specific benchmarks; and (2) others resort to injecting excessive auxiliary prompts into general-purpose models that lack fine-tuning or the capability for deep misinformation reasoning. Large Reasoning Models (LRMs), such as Qwen3 [41, 42] and DeepSeek [14], represent the forefront of AI with advanced reasoning, deep chain-of-thought processes, and efficient knowledge utilization [50], yet their use remains largely confined to text-only domains. While recent open-source efforts have advanced deep reasoning on structured tasks like academic QA and single-image inference [13, 11], their application to misinformation detection remains underexplored. Future work should focus on equipping MLLMs with autonomous reasoning and long-chain-of-thought capabilities tailored to the complexities of misinformation detection—a critical yet challenging direction.

To address these challenges, we introduce FakeVV, the largest and most comprehensive dataset for video misinformation detection to date. It offers extensive topical coverage, a wide temporal range (2006–2025), and enriched annotations. We begin by collecting 100k high-quality video-text pairs from four official news channels on Internet, preserving associated metadata such as user comments, engagement metrics (*e.g.*, likes), and timestamps. Given the frequent absence of video captions in the news domain, we develop a GPT-4o-based caption generation pipeline, guided by video titles and named entities, to generate high-quality captions that are essential for the subsequent training pipeline. To construct challenging misinformation samples, we propose a non-random entity replacement strategy that introduces semantically inconsistent video-text pairs. Leveraging CLIP-based similarity matching on video content, titles, and cross-modal representations, we retrieve similar news samples and replace entities of four key types—persons, locations, events, and organizations—to fabricate realistic fake videos. Each fabricated sample is annotated with the manipulated entity, thereby supporting fine-grained reasoning supervision and interpretable evaluation.

Building upon the FakeVV dataset, we further propose Fact-R1, the first multimodal misinformation detection framework that integrates deep reasoning with collaborativ rule-based reinforcement learning. Fact-R1 is trained via a three-stage optimization pipeline tailored for misinformation scenarios: (1) misinformation Long Chain-of-Thought Instruction Tuning: We construct 85K CoT training samples using DeepSeek-R1 [14] guided by prompts derived from the generated video captions. The outputs are filtered based on answer correctness and whether the reasoning correctly identifies the manipulated entity. This stage equips the model with the capability to perform deep, autonomous, and long-chain reasoning in the domain of misinformation. (2) Preference Alignment via Direct Preference Optimization (DPO) [28]: A curated 5K DPO dataset with human preference labels is used to refine the model's reasoning coherence and factual accuracy, addressing issues such as hallucinations, incorrect fake-entity attribution, and answer formatting inconsistencies. (3) Group-Relative Policy Optimization (GRPO) [32] with a Verifiable Reward Function: In the final stage, we define a task-specific, rule-based misinformation reward function to evaluate both reasoning trajectories and entity-level correctness over 15K new samples and 5K auxiliary-task samples. To further enhance visual reasoning, the auxiliary tasks include News Image OCR and News Video Caption, which are designed to improve visual-textual alignment and guide the model toward better interpretation of manipulated content. This encourages the model to explore diverse and verifiable reasoning strategies beyond surface-level patterns and adaptively optimize for misinformation detection tasks.

Overall, our main contributions can be summarized as follows: (1) We construct FakeVV, the largest and most comprehensively annotated news-domain video misinformation dataset. It features high-quality, generated video captions specifically designed to support misinformation reasoning tasks. (2) We propose a novel non-random entity replacement strategy that synthesizes semantically inconsistent misinformation samples, enabling explainable evaluation through fine-grained manipulated-entity annotations, supporting interpretable model behavior. (3) We introduce Fact-R1, the first multimodal misinformation detection model that combines deep reasoning with collaborative rule-based reinforcement learning. Fact-R1 establishes a new paradigm for misinformation detection by bridging video understanding, reasoning alignment, and verifiable explainability. (4) We design a task-specific reward function for misinformation detection and introduce a novel reinforcement learning framework with multiple auxiliary tasks to enhance multimodal reasoning.

## 2 Related Works

**Video Misinformation Detection.** Existing approaches for video misinformation detection primarily focus on feature extraction and integration across multiple modalities, including linguistic patterns [29], emotional acoustic cues [15], and multimodal inconsistencies [9]. Tarhouni *et al.* [34] leveraged audio and frame watermarks for cross-channel manipulation detection, while You *et al.* [44] proposed a fusion model combining topic information and keyframe features. Qi *et al.* [24] explored cross-modal correlations with social context and further introduced the NEED framework [26] for neighborhood relationship modeling. Zong *et al.* [53] recently integrated large language models (LLMs) with a viewpoint evolution mechanism to support cross-modal reasoning. To facilitate this research, several dedicated datasets have been developed. Papadopoulou *et al.* [22], Hou *et al.* [15], and Shang *et al.* [30] collected data on multilingual news, prostate cancer misinformation, and COVID-19-related content, respectively. However, these datasets often suffer from limited scale, narrow topical coverage, or lack of public availability. Later works such as FakeSV [24] and FakeTT [6] address some of these issues but still face challenges in data diversity, temporal range, and interpretability—key factors for developing robust and generalizable misinformation detection models. Besides, current methods remain limited in reasoning capability, restricting their ability to interpret complex misinformation patterns and generalize across diverse manipulation strategies.

**Reasoning in Multimodal LLMs.** The rapid progress of large language models (LLMs), such as OpenAI o1 [16] and DeepSeek R1 [14], has fueled growing interest in complex reasoning tasks, while multimodal large language models (MLLMs) aim to extend this capability into the visual domain for cross-modal applications [37]. Early work often relies on neuro-symbolic reasoning frameworks [38], such as the Differentiable Logical Formalism for VQA introduced by Amizadeh *et al.* [2]. Recent MLLMs further improve visual reasoning through techniques like Visual Chain-of-Thought [31, 49], enhanced search strategies [40], reasoning transfer via data organization [12], and image-based reasoning trajectories [17], contributing to advances in both performance and interpretability. In this work, we adapt these reasoning capabilities to the task of video misinformation detection, aiming to uncover deceptive patterns in multimodal content.

Table 1: Summary of datasets of video detection. Metadata refers to basic statistics such as # of likes/stars/edit time. "-" represents the exact time range is not found in the paper.

| Dataset | Video | Title | Metadata | Comment | #Fake | #Real | Time Range | Interpretability | Construction Mode |
|---|---|---|---|---|---|---|---|---|---|
| FVC [22] | ✓ | ✓ | ✓ | ✓ | 2,916 | 2090 | - | ✗ | Web collection |
| VAVD [21] | ✓ | ✓ | ✓ | ✓ | 123 | 423 | 2013/09-2016/10 | ✗ | Web collection |
| YouTube-Covid [29] | ✓ | ✓ | ✗ | ✓ | 113 | 67 | 2019/10-2020/04 | ✗ | Web collection |
| TikTok-Covid [30] | ✓ | ✓ | ✗ | ✗ | 226 | 665 | - | ✗ | Web collection |
| TSC [43] | ✓ | ✓ | ✓ | ✓ | 262 | 383 | - | ✗ | Web collection |
| MYVC [9] | ✓ | ✓ | ✗ | ✗ | 902 | 903 | - | ✗ | Web collection |
| FakeSV [24] | ✓ | ✓ | ✓ | ✓ | 1,827 | 1,827 | 2017/10-2022/02 | ✗ | Web collection |
| FakeTT [6] | ✓ | ✓ | ✓ | ✓ | 1,172 | 819 | 2019/05-2024/03 | ✗ | Web collection |
| **FakeVV (ours)** | ✓ | ✓ | ✓ | ✓ | **51,000** | **51,000** | **2006/11-2025/02** | ✓ | **Autotectonics** |

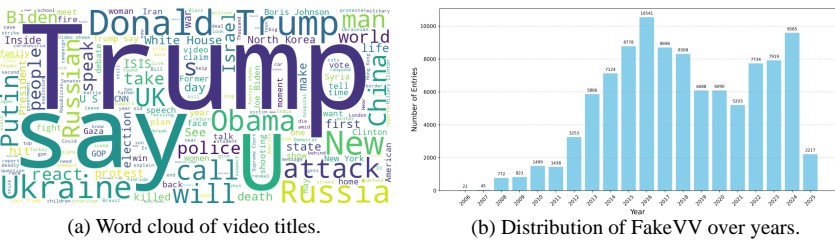

(a) Word cloud of video titles.  (b) Distribution of FakeVV over years.

Figure 2: The statistics of FakeVV dataset.

## 3  FakeVV Dataset Construction

In this section, we provide a detailed description of the construction process of the FakeVV dataset, as shown in Figure 3, which consists of the following components.

**Data Collection.** To ensure diversity and relevance, we collect the most popular videos from four official news accounts—BBC News, Guardian News, CNN, and The New York Times—covering November 2006 to February 2025, with approximately 120,000 raw samples. We filter out non-news content, exclude videos over five minutes, and remove near-duplicate events by clustering semantic representations of video titles, resulting in 100,000 curated news videos. We also collect metadata such as related articles, tags, timestamps, and user comments to support future research.

**Data Pre-processing.** To generate enriched video captions that provide detailed visual information for chain-of-thought reasoning, we propose a news-domain video captioning method. Unlike conventional approaches, our method produces captions that incorporate not only basic visual descriptions but also specific entities and semantically rich content tailored to news contexts. These enhanced captions help large language models (LLMs) achieve a deeper understanding of video news, supporting the generation of higher-quality reasoning chains. Our approach builds on Qwen-VL's [3] video processing strategy by first extracting 16 keyframes from each news video and identifying visual entities using the Google Vision API. We then employ GPT-4o [1] to generate instruction-based captions for these keyframes, guided by prompts that include auxiliary information such as the original news title and relevant entity names. Finally, we apply rigorous post-processing and validation to ensure that the enriched captions accurately reflect the core information of the news videos. Detailed data generation procedures can be found in the supplementary materials.

**Misinformation Data Construction.** To construct fine-grained and semantically inconsistent misinformation samples, we propose a non-random entity matching strategy designed to introduce challenging yet realistic perturbations. Specifically, we generate four types of forged news samples by replacing different types of entities associated with similar news content: person-name inconsistency, location inconsistency, event inconsistency, and organization-name inconsistency.

Each news sample is represented as a tuple (*Img*, *Title*), where *Img* denotes the first keyframe of the original news video, and *Title* refers to its corresponding news title. First, we utilize the CLIP [27] model to extract the visual and textual semantic representations of the news samples, forming a news semantic repository. Subsequently, any given news sample is used as a query, and based on the cosine similarity of the representations, the top-3 most semantically similar candidate samples are retrieved from the repository to serve as potential sources for entity substitution. The retrieval

process randomly selects one of the following five strategies: (1) Visual-to-Visual retrieval; (2) Visual-to-Textual retrieval; (3) Textual-to-Visual retrieval; (4) Textual-to-Textual retrieval; (5) Random selection. Using the retrieved three candidate samples, we input both the query sample and the candidate samples into the GPT-4o model. Through carefully designed prompts, a randomly selected target entity in the news headline of the query sample is replaced, thereby generating a forged news title that preserves surface fluency but violates factual consistency. In addition to the forged titles, we also prompt the model to produce corresponding metadata, including the manipulated fake entity and its type. Using this methodology, we generated 102,000 forged news samples, each containing the news video, title, forged entity, and manipulation type. The dataset was partitioned into 100,000 training samples (before 2025) and 2,000 testing samples (2025), enabling robust evaluation of the model's generalization to unseen data.

**Data Analysis.** We present a comprehensive comparison between the FakeVV dataset and existing video misinformation detection datasets, as summarized in Table 1. FakeVV is an order of magnitude larger than prior datasets, with a balanced distribution (50% authentic, 50% manipulated) and no significant unimodal textual bias. Crucially, it provides fine-grained annotations of forged entities as explainability labels, along with manipulation types, offering explicit and interpretable evaluation signals. These features support rigorous evaluation of model reasoning, including the use of LLMs like GPT-4o and human evaluators to verify the accuracy of inferred manipulations. Moreover, all manipulated samples in FakeVV are derived from authentic videos, avoiding artifacts commonly introduced by web-scraped data. Figure 2 illustrates its thematic and temporal distribution. Thematically, the dataset focuses on political and international affairs, with frequent keywords such as "Trump," "Russia," and "Ukraine," covering high-impact misinformation topics. Temporally, FakeVV spans from November 2006 to February 2025, with notable peaks in 2016 and 2024—periods marked by major global events like U.S. presidential elections and the rise of social media-driven news.

# 4 Method

As illustrated in Figure 3, the training pipeline of Fact-R1 consists of three sequential stages designed to progressively endow the model with the ability to handle complex misinformation detection tasks.

## 4.1 Long-CoT Instruction Tuning

Existing open-source MLLMs exhibit limited reasoning capabilities in the misinformation domain. To address this, we construct a misinformation long-Chain-of-Thought (long-CoT) instruction tuning dataset from the FakeVV corpus and fine-tune an open-source MLLM to inject domain-specific reasoning knowledge and long-CoT capability, providing a foundation for subsequent high-level reasoning tasks. As shown in Figure 3, the long-CoT dataset is built using previously generated news-domain video captions and audio transcripts as textual proxies for video inputs, aligning with the text-only optimization of current deep reasoning models such as DeepSeek-R1. For each instance, the video title, caption, and audio are provided to DeepSeek-R1, which assesses the veracity and generates a multi-step reasoning trace enclosed within `<think>` tags. We filter the generated responses by retaining only samples that meet two criteria: (1) the model correctly predicts the real/fake label, and (2) the reasoning trace explicitly and accurately identifies the manipulated entity. This process yields approximately 85,000 high-quality long-CoT instances for the first-stage instruction tuning.

## 4.2 Preference Alignment via DPO

To further align the model's outputs with human expectations, as shown in Figure 3, we fine-tune Fact-R1 using Direct Preference Optimization (DPO) on a curated set of 5k human-annotated preference pairs. These samples specifically target common failure cases, such as inaccurate answers, hallucinated fake entity descriptions, and commonsense inconsistencies. Each pair consists of a human-preferred output $y_{\text{label}}$ and a suboptimal model-generated response $y_{\text{pred}}$, where the preferred output demonstrates superior factual accuracy, reasoning coherence, and output formatting. Following the DPO objective [28], the model is optimized to assign higher likelihood to $y_{\text{label}}$ without requiring an explicit reward model. The loss is defined as:

$$\mathcal{L}_{\text{DPO}} = -\log \frac{\exp\left(\beta \cdot r(y_{\text{label}})\right)}{\exp\left(\beta \cdot r(y_{\text{label}})\right) + \exp\left(\beta \cdot r(y_{\text{pred}})\right)}, \tag{1}$$

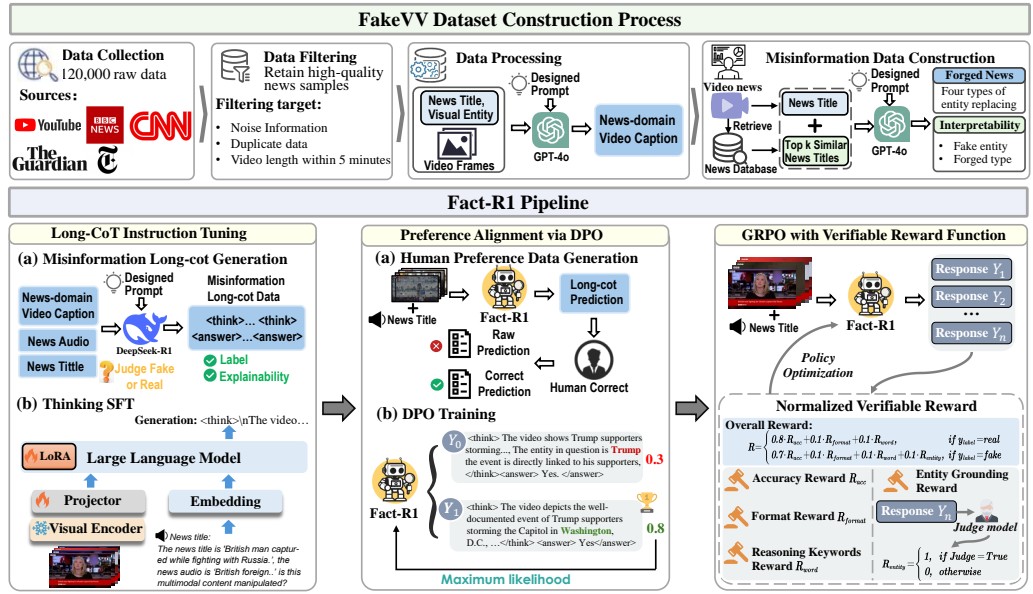

Figure 3: The overall architecture of the Fact-R1 is illustrated, with the upper part showing the FakeVV dataset construction process and the lower part presenting the training pipeline of Fact-R1.

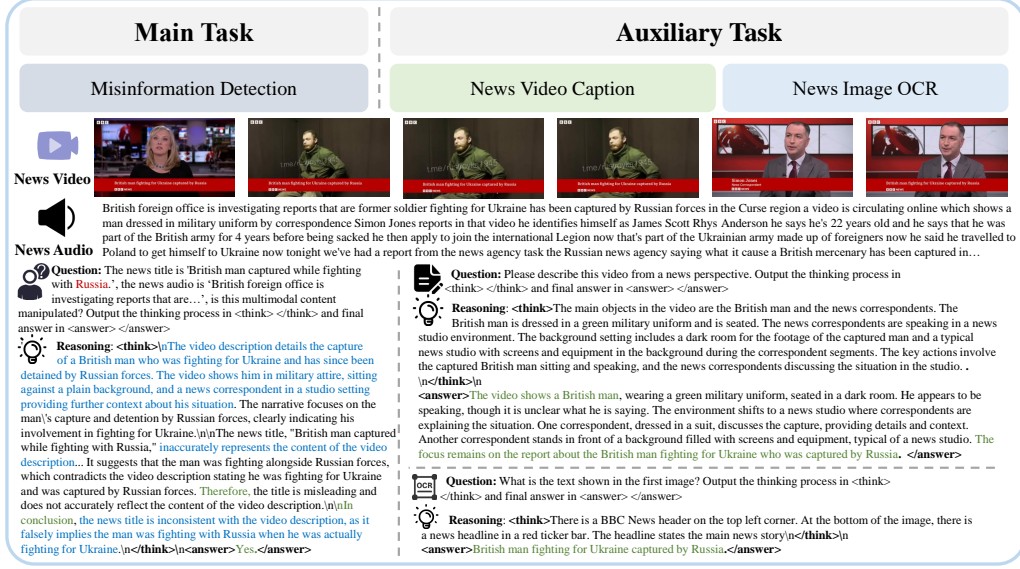

Figure 4: Fact-R1 incorporates News Video Caption and News Image OCR as auxiliary tasks to enhance its misinformation detection capability.

where $r(y)$ denotes the reward computed from the log-ratio of likelihoods under the current policy and a frozen reference policy. This preference-based alignment encourages Fact-R1 to generate outputs that better match human expectations in reasoning accuracy and response structure, while maintaining stable training.

## 4.3 GRPO with Verifiable Reward Function

Inspired by recent advances in purely reinforcement-based training [14], we explore the potential of MLLMs to acquire sophisticated reasoning abilities through self-evolution driven solely by reward-guided optimization, without reliance on supervised instruction.

Table 2: Performance comparison on three real-world datasets. The best results are in red bold.

| Model | FakeSV | | | | FakeTT | | | | FakeVV | | | |
|---|---|---|---|---|---|---|---|---|---|---|---|---|
| | Acc | Prec | Rec | F1 | Acc | Prec | Rec | F1 | Acc | Prec | Rec | F1 |
| BERT [10] | 65.4 | 66.0 | 66.5 | 66.2 | 68.7 | 67.5 | 67.5 | 67.5 | 60.4 | 57.9 | 56.8 | 57.3 |
| TikTec [30] | 64.8 | 63.2 | 61.9 | 62.5 | 61.1 | 64.8 | 64.2 | 64.5 | 59.3 | 59.1 | 59.5 | 59.3 |
| FANVM [9] | 65.4 | 66.1 | 64.3 | 65.2 | 68.9 | 64.7 | 68.8 | 67.1 | 61.9 | 60.7 | 60.8 | 60.8 |
| SV-FEND [24] | 67.1 | 67.4 | 66.3 | 66.8 | 67.6 | 72.2 | 69.0 | 70.6 | 70.9 | 71.4 | 71.3 | 71.3 |
| FakingRec [6] | 69.5 | 69.7 | 70.4 | 70.0 | 71.0 | 71.9 | 72.0 | 72.0 | 72.1 | 72.4 | 71.6 | 72.0 |
| Gemini2-thinking [35] | 63.1 | 61.8 | 61.9 | 61.9 | 56.6 | 55.2 | 55.3 | 55.3 | 51.5 | 46.0 | 46.0 | 48.6 |
| GPT-4o [1] | 66.6 | 65.2 | 64.7 | 64.9 | 57.9 | 57.8 | 62.9 | 63.7 | 56.0 | 60.4 | 35.0 | 44.3 |
| GPT-o1-mini [16] | 60.3 | 57.7 | 56.5 | 57.1 | 52.5 | 51.6 | 51.7 | 51.7 | 47.5 | 46.9 | 37.6 | 41.8 |
| DeepSeek-R1 [14] | 61.8 | 60.4 | 60.3 | 60.3 | 49.8 | 52.6 | 52.5 | 52.6 | 53.5 | 58.1 | 25.2 | 35.1 |
| Qwen2.5-VL-7B [3] | 55.6 | 55.5 | 55.7 | 55.6 | 54.9 | 54.0 | 54.1 | 54.0 | 52.9 | 51.1 | 51.1 | 51.1 |
| Qwen2.5-VL-72B [3] | 57.6 | 55.4 | 55.2 | 55.3 | 59.2 | 58.1 | 58.3 | 58.2 | 54.0 | 60.0 | 24.0 | 34.3 |
| QVQ-72B-preview [36] | 60.8 | 59.0 | 58.8 | 58.9 | 58.1 | 54.0 | 52.8 | 53.4 | 53.5 | 52.6 | 52.6 | 52.6 |
| InternVL2.5-8B [8] | 49.8 | 52.6 | 52.5 | 52.6 | 43.9 | 44.0 | 44.0 | 44.0 | 53.5 | 58.5 | 24.0 | 34.0 |
| InternVL2.5-78B-MPO [39] | 57.5 | 53.0 | 52.0 | 52.5 | 59.2 | 57.1 | 56.7 | 56.9 | 54.0 | 60.0 | 24.0 | 34.3 |
| **Fact-R1** | **75.6** | **77.7** | **72.0** | **74.7** | **74.4** | **77.8** | **68.3** | **72.7** | **81.2** | **84.5** | **76.4** | **80.3** |

Table 3: Ablation study on the contribution of key components in Fact-R1.

| Variant | FakeTT | | FakeVV | |
|---|---|---|---|---|
| | ACC | F1 | ACC | F1 |
| w/o SFT | 70.9 | 71.7 | 66.8 | 66.8 |
| w/o DPO | 72.1 | 72.4 | 80.4 | 79.9 |
| w/o GRPO | 70.7 | 70.6 | 79.8 | 79.1 |
| w/o Audio | 73.0 | 72.3 | 79.0 | 77.7 |
| **Fact-R1** | **74.4** | **72.7** | **81.2** | **80.3** |

Table 4: Evaluating the Impact of the Reward Function in Fact-R1.

| Variant | FakeTT | | FakeVV | |
|---|---|---|---|---|
| | ACC | F1 | ACC | F1 |
| w/o Keywords | 71.1 | 72.0 | 78.6 | 79.9 |
| w/o Entity | 70.4 | 71.4 | 79.4 | 80.0 |
| w/o Ocr | 71.9 | 71.8 | 80.8 | 80.2 |
| w/o Caption | 71.6 | 71.6 | 75.5 | 77.7 |
| **Fact-R1** | **74.4** | **72.7** | **81.2** | **80.3** |

Figure 5: The interpretability accuracy of the outputs from the six models.

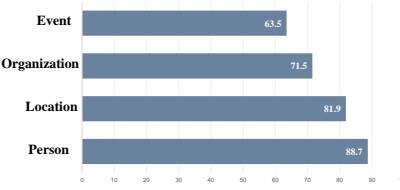

Figure 6: Interpretability score distribution across forgery types.

**Group Relative Policy Optimization (GRPO).** We adopt GRPO [32] to fine-tune the policy network without requiring an explicit value function or critic model. GRPO estimates relative output quality by comparing a group of candidate responses generated under the current policy. Given an input, the model samples a set of responses $\{y_1, y_2, \ldots, y_G\}$, each assigned a reward based on correctness and alignment with human preferences. The normalized advantage for each response $y_i$ is computed as:

$$A_i = \frac{r_i - \text{mean}(\{r_1, \ldots, r_G\})}{\text{std}(\{r_1, \ldots, r_G\})}. \tag{2}$$

This normalization mitigates reward variance and stabilizes training. The policy is optimized to maximize the importance-weighted advantage while regularizing deviation from a reference model:

$$\mathcal{L}_{\text{GRPO}} = \frac{1}{G} \sum_{i=1}^{G} \left( \min \left( \frac{\pi_\theta(y_i)}{\pi_{\text{old}}(y_i)} A_i, \ \text{clip} \left( \frac{\pi_\theta(y_i)}{\pi_{\text{old}}(y_i)}, 1 - \epsilon, 1 + \epsilon \right) A_i \right) - \beta \mathbb{D}_{\text{KL}}(\pi_\theta \parallel \pi_{\text{ref}}) \right). \tag{3}$$

Here, $\pi_\theta$ is the current policy, $\pi_{\text{old}}$ the sampling policy, and $\pi_{\text{ref}}$ a fixed reference model.

**Verifiable Reward Modeling for Misinformation Detection** To effectively adapt Reinforcement Learning (RL) to the misinformation domain, we design a rule-based verifiable reward function for video misinformation detection. The total reward $R$ is composed of four components:

1. Accuracy Reward ($R_{\mathrm{acc}}$): Measures whether the model's predicted class matches the ground truth.
2. Format Reward ($R_{\mathrm{format}}$): Encourages proper formatting of reasoning content within the `<think>` and `</think>` tags, and the final answer within the `<answer>` and `</answer>` tags.
3. Reasoning Keywords Reward ($R_{\mathrm{word}}$): Rewards reasoning traces that contain key reflective terms such as "First", "However", or "In conclusion".
4. Entity Grounding Reward ($R_{\mathrm{entity}}$): Ensures the model correctly identifies manipulated entities in the reasoning trace. A judge model $J$ is prompted to verify whether the predicted reasoning (within the `<think>` tag) contains a correct reference to the ground-truth fake entity $e^*$:

$$R_{\mathrm{entity}}(q, y) = \begin{cases} 1, & \text{if } J(y_{\texttt{<think>}}, e^*) = \texttt{True} \\ 0, & \text{otherwise} \end{cases} \tag{4}$$

Finally, the overall verifiable reward function $R$ is defined as:

$$R = \begin{cases} 0.8 \cdot R_{\mathrm{acc}} + 0.1 \cdot R_{\mathrm{format}} + 0.1 \cdot R_{\mathrm{word}}, & \text{if } y_{\mathrm{label}} = \texttt{real} \\ 0.7 \cdot R_{\mathrm{acc}} + 0.1 \cdot R_{\mathrm{format}} + 0.1 \cdot R_{\mathrm{word}} + 0.1 \cdot R_{\mathrm{entity}}, & \text{if } y_{\mathrm{label}} = \texttt{fake} \end{cases} \tag{5}$$

As illustrated in Figure 4, we introduce News Video Caption and News Image OCR as auxiliary tasks for misinformation detection, jointly trained to enhance the visual reasoning capability of Fact-R1. These auxiliary tasks explicitly guide the model to learn fine-grained visual-textual alignment, thereby improving its perception and understanding of manipulated content. The reward function for each auxiliary task consists of two components: Accuracy Reward and Format Reward, weighted at $0.9$ and $0.1$, respectively. The accuracy reward $R_{\mathrm{acc}}$ is computed using task-specific evaluation metrics:

$$R_{\mathrm{acc}}(q, y) = \begin{cases} 1 - \frac{\mathrm{EditDistance}(q, y)}{\max(|q|, |y|)}, & \text{if task} = \text{OCR}, \\ \text{ROUGE-L}(q, y), & \text{if task} = \text{Caption}. \end{cases} \tag{6}$$

where $q$ denotes the predicted sequence and $y$ is the ground truth. For the OCR task, we adopt normalized edit distance to measure character-level accuracy, while for the caption task, we use the ROUGE-L score normalized to the $[0, 1]$ range to evaluate sequence similarity. By incorporating these auxiliary reward functions into the reinforcement learning process, Fact-R1 achieves more stable training and effectively integrates OCR and caption-based reasoning, thereby enhancing its ability to detect and interpret video-based misinformation through stronger visual-textual alignment.

## 5 Experiments

### 5.1 Experimental Settings

**Dataset.** For evaluation, we use the FakeVV testing set along with two widely adopted benchmarks. FakeSV [24] is a large-scale Chinese short video dataset for fake news detection, enriched with social context features such as user comments and publisher metadata; video titles are translated into English for compatibility. FakeTT [6] is an English-language dataset from TikTok, covering a wide range of misinformation topics. Following [6], we adopt a temporal split by using the most recent 15% of samples from FakeSV and FakeTT as testing sets. All baseline models are trained on the FakeVV, FakeTT, and FakeSV training sets, which provide balanced datasets with 53,800 real and 53,800 fake video news samples each. We report four standard evaluation metrics: Accuracy (ACC), Precision, Recall, and F1 Score, providing a comprehensive view of classification effectiveness.
**Baselines.** We compare Fact-R1 against 15 competitive baselines in four groups: (1) single-modal methods including BERT [10]; (2) multimodal methods including TikTec [30], FANVM [9], SV-FEND [24], and FakingRec [6]; (3) closed-source MLLMs including Gemini2-thinking [35], GPT-4o [1], and GPT-o1-mini [16]; (4) open-source MLLMs including Qwen2.5-VL [3], InternVL2.5 [8], QVQ-72B [36], InternVL2.5-MPO [39] and DeepSeek-R1 [14]. The first two groups are trained using the FakeVV training set, while the latter two are evaluated in a zero-shot setting. For models without native video support (*e.g.*, DeepSeek-R1, GPT-o1-mini), we use news-domain video captions as textual surrogates. Other closed-source or inaccessible methods are excluded from the evaluation.
**Implementation Details.** All experiments are conducted using PyTorch on 8×NVIDIA A100 GPUs,

with Qwen2.5-VL as the base MLLM and the default 8-frame sampling strategy. Stage 1 (Long-CoT Tuning). We freeze the visual encoder and apply LoRA ($r = 128$, $\alpha = 256$), using learning rates of $2 \times 10^{-5}$ (LoRA) and $2 \times 10^{-6}$ (multimodal projector). Training is performed for 2 epochs with batch size 2, using AdamW optimizer. Stage 2 (DPO). We train for 1 epoch on 5k human preference pairs with batch size 4 and a learning rate of $2 \times 10^{-5}$. The reference model is frozen, and gradient clipping is applied with a max norm of 1.0. Stage 3 (GRPO). We train for 172 steps using the verifiable reward function, sampling 5 candidate responses per input and applying a learning rate of $1 \times 10^{-6}$. The speech recognition system is employed to transcribe audio recordings into text.

## 5.2 Comparison to State-of-the-Art Approaches

As shown in Table 2, the proposed Fact-R1 achieves the best overall performance in video misinformation detection. Among baselines, BERT performs well, suggesting the strength of textual signals, while FakingRec benefits from modeling video production semantics. SV-FEND also performs competitively by leveraging multi-modal cues via attention mechanisms. For closed-source MLLMs, GPT-4o achieves the highest accuracy, with Gemini2-thinking following closely. Notably, text-only models like DeepSeek-R1 and GPT-o1-mini show strong results, highlighting the power of high-level reasoning. Among open-source models, QVQ-72B leads, while InternVL2.5-MPO and Qwen2.5-VL-7B show promising but varied performance. In contrast, InternVL2.5-8B consistently underperforms. Overall, larger models tend to perform better, though performance varies across datasets, reflecting differing task difficulty. The generally weaker results of non-finetuned models further emphasize the need for reasoning-oriented training. These results demonstrate that Fact-R1's superior performance arises from its tailored misinformation reasoning design, combining long-CoT instruction tuning, DPO-based preference alignment, and GRPO-driven policy optimization.

## 5.3 Explainability Analysis

Due to the lack of annotated manipulation process information in existing misinformation detection datasets, current explainability efforts suffer from inconsistent evaluation standards, which significantly hinders the development of trustworthy misinformation detection models. During evaluation, we retain both the reasoning traces generated by each model and the ground-truth manipulated entity (i.e., *fake entity*) for each fake sample. GPT-4o-mini is employed as a judge model to assess whether the fake entity is correctly identified and described within the reasoning process. Accuracy is used as the evaluation metric, with the prompt design detailed in the supplementary material. To decouple detection performance from reasoning quality, we only evaluate samples in which the model correctly predicts the fake label. As shown in Figure 5, Fact-R1 demonstrates strong reasoning ability by consistently describing the correct fake entities rather than overfitting to specific patterns in the dataset. Figure 6 shows that describing manipulated *Event* and *Organization* types remains more challenging than other categories.

## 5.4 Ablation Study

We conduct ablation experiments to verify the contribution of each stage in our proposed three-stage training pipeline, as shown in Table 3. *w/o SFT* denotes Fact-R1 without misinformation long-Chain-of-Thought (CoT) instruction tuning, *w/o DPO* removes the preference alignment stage using Direct Preference Optimization, *w/o GRPO* removes Group Relative Policy Optimization with the verifiable reward function. Our observations confirm that the staged training design plays a crucial role in progressively building up reasoning capability. Table 4 presents the effectiveness of our designed reward function and the two auxiliary tasks. Here, *w/o Keywords* and *w/o Entity* indicate the removal of the Reasoning Keywords Reward and Entity Grounding Reward, respectively, while *w/o Ocr* and *w/o Caption* denote the exclusion of the News Video Caption and News Image OCR auxiliary tasks. The results demonstrate that these components significantly contribute to enhancing the reasoning ability and overall performance of Fact-R1.

# 6 Conclusion

In this work, we address the underexplored challenge of video misinformation detection by introducing FakeVV, a large-scale, diverse benchmark with fine-grained and interpretable annotations. To

effectively leverage this resource, we propose Fact-R1, a novel multimodal framework that unifies deep reasoning with collaborative rule-based reinforcement learning. Fact-R1 is trained through a three-stage pipeline, enabling the model to generate structured, explainable reasoning traces. Our approach not only advances the state of the art in multimodal misinformation detection, but also establishes a scalable, interpretable paradigm for aligning large multimodal models with complex real-world reasoning tasks. Comprehensive experiments validate the effectiveness of Fact-R1.[4]

# 7  Limitations

In this work, we propose Fact-R1, a deep reasoning model for video misinformation detection, which achieves state-of-the-art performance on three benchmark datasets. Despite these promising results, the task remains inherently challenging and fraught with limitations.

First, the accuracy of misinformation detection remains generally low in real-world scenarios due to the complexity of video content, the ambiguity of factual claims, and the constantly evolving nature of misinformation strategies. While our model demonstrates strong performance on curated datasets, its generalizability and robustness in open-domain or adversarial environments require further investigation.

Finally, over-reliance on automated fact-checking systems carries significant societal risks. Misclassifying authentic content as misinformation can suppress legitimate discourse and harm the reputations or economic viability of content creators. Conversely, undetected misinformation—especially when reinforced by flawed but confident reasoning—may erode public trust, amplify polarization, and destabilize information ecosystems. We acknowledge that our dataset, sourced exclusively from four Western news outlets, may introduce geographic and cultural biases; accordingly, the generalizability of our findings to other regions and contexts remains an open question.

# 8  Societal Impacts

**Positive Societal Impacts**

Our work aims to enhance the reliability and transparency of misinformation detection, particularly in the context of video content, which is increasingly prevalent and impactful in public discourse. By equipping multimodal language models with structured reasoning capabilities, Fact-R1 can serve as an effective assistive tool for human fact-checkers, enabling more informed decision-making in combating misinformation. The structured outputs may also contribute to media literacy by offering interpretable justifications, thereby supporting responsible content consumption and improving public trust in automated fact-checking systems.

**Negative Societal Impacts**

Despite its intended benefits, our approach may carry potential risks if deployed without proper safeguards. Incorrect predictions—especially when accompanied by seemingly coherent but factually incorrect reasoning—could lead to misclassification of legitimate content or failure to detect harmful misinformation. Over-reliance on automated systems may also reduce critical human oversight and unintentionally suppress diverse viewpoints. Moreover, hallucinated reasoning traces from large language models, if not clearly flagged, might mislead both users and reviewers in high-stakes scenarios.

# Acknowledgment

This work was supported by the National Natural Science Foundation of China (NSFC) under Grant 62476260, 62225207 and 62436008, the Fundamental Research Funds for the Central Universities under Grant WK2100000057.

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

Figure 7: An example from the dataset to illustrate the News-domain Video Caption Generation.

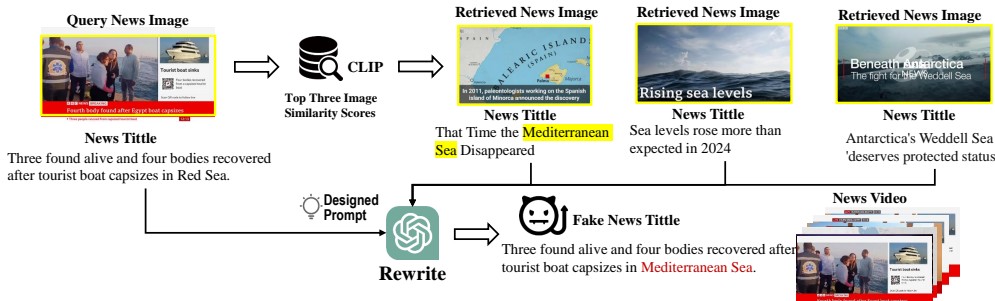

Figure 8: The candidate news sample retrieval pipeline based on Visual-to-Visual matching employed in the FakeVV dataset.

## A    Detailed Experimental Settings

### A.1    Datasets

To evaluate the performance of our proposed framework Fact-R1 and several baseline models, we conduct experiments on three real-world short video misinformation datasets: FakeSV, FakeTT, and FakeVV. The statistics and key characteristics of these datasets are summarized in Table 1. Following the original setup in [24], we adopt a chronological split with a ratio of 70% for training, 15% for validation, and 15% for testing, simulating realistic scenarios where only past data is available for detecting future misinformation.

The details of each dataset are as follows:

**FakeSV**: A dataset for Chinese short video misinformation detection, constructed by scraping news videos from Chinese short video platforms and enriching them with social context features such as user comments and publisher metadata.

**FakeTT**: A curated English short video misinformation dataset, focusing on misinformation detection on platforms like TikTok. Each sample includes the video content, title, and associated metadata, providing diverse and realistic misinformation scenarios from widely-used social media environments.

**FakeVV**: Our proposed large-scale English video misinformation benchmark, featuring fine-grained forged entity annotations, manipulation types, and explainability labels.

### A.2    Baselines

To verify the effectiveness of Fact-R1, we compare it against 15 competitive open-source accessible baselines across four groups. Other closed-source or inaccessible methods are excluded from the evaluation.

**Algorithm 1:** GRPO with Task-Specific Reward Functions

---

**Input:** Mini-batch $\{(x_i, q_i, t_i)\}_{i=1}^{N}$, where task flag $t_i \in \{\textbf{MD}, \textbf{OCR}, \textbf{CAP}\}$;
MD = Misinformation Detection, OCR = Image OCR, CAP = Video Caption;
current policy $\pi_\theta$, reference policy $\pi_{\text{ref}}$, group size $G$.
**Output:** Updated policy $\pi_\theta$.

1 **for** *each sample* $(x_i, q_i, t_i)$ **do**
2      **(1) Sample** $G$ candidate outputs: $\{y_{i1}, \ldots, y_{iG}\} \sim \pi_\theta(\cdot \mid x_i, q_i)$;
3      **(2) Compute task-specific reward** $R$;
4      **for** $k = 1$ **to** $G$ **do**
5          **if** $t_i = \textbf{MD}$ **then**
6              $R_{\text{acc}} \leftarrow \mathbb{1}[y_{ik}^{\text{label}} = y_i^{\text{gt}}]$;
7              $R_{\text{format}} \leftarrow \mathbb{1}[\text{tags correct}]$;
8              $R_{\text{word}} \leftarrow \mathbb{1}[\text{key words}]$;
9              $R_{\text{entity}} \leftarrow \begin{cases} J(y_{ik, \texttt{<think>}}, e_i^*), & y_{ik}^{\text{label}} = \texttt{fake}, \\ 0, & \text{otherwise.} \end{cases}$;
10

$$R = \begin{cases} 0.7 R_{\text{acc}} + 0.1 R_{\text{format}} + 0.1 R_{\text{word}} + 0.1 R_{\text{entity}}, & \text{if } y_{ik}^{\text{label}} = \texttt{fake}, \\ 0.8 R_{\text{acc}} + 0.1 R_{\text{format}} + 0.1 R_{\text{word}}, & \text{if } y_{ik}^{\text{label}} = \texttt{real}. \end{cases}$$

11          **else if** $t_i = \textbf{OCR}$ **then**
12              $R_{\text{acc}} = 1 - \dfrac{\text{EditDist}(y_{ik}, y_i^{\text{gt}})}{\max(|y_{ik}|, |y_i^{\text{gt}}|)}$;
13              $R_{\text{format}} \leftarrow \mathbb{1}[\text{tags correct}]$;
14              $R = 0.9 R_{\text{acc}} + 0.1 R_{\text{format}}$;
15          **else**
16              $R_{\text{acc}} = \text{ROUGE-L}(y_{ik}, y_i^{\text{gt}})$;
17              $R_{\text{format}} \leftarrow \mathbb{1}[\text{tags correct}]$;
18              $R = 0.9 R_{\text{acc}} + 0.1 R_{\text{format}}$;

19      **(3) Normalize** rewards: $A_i = \dfrac{R - \mu_i}{\sigma_i}$,    $\mu_i, \sigma_i$ are the mean/std of $\{R_1, \ldots, R_G\}$;
20      **(4) GRPO loss**:

$$\mathcal{L}_i = \frac{1}{G} \sum_{k=1}^{G} \left( \min\left( \tfrac{\pi_\theta}{\pi_{\text{old}}} A_i, \text{clip}\left( \tfrac{\pi_\theta}{\pi_{\text{old}}}, 1 - \epsilon, 1 + \epsilon \right) A_i \right) - \beta \, \text{D}_{\text{KL}}(\pi_\theta \parallel \pi_{\text{ref}}) \right)$$

     ;
21 **Update** parameters: $\theta \leftarrow \theta - \eta \, \nabla_\theta \sum_i \mathcal{L}_i$;
22 **return** $\pi_\theta$

---

**(1) Single-modal detection methods**:

- **BERT** [10]: A widely used pre-trained language model designed for natural language understanding tasks, employed here for text-based misinformation detection using video titles and descriptions.

**(2) Multimodal detection methods**:

- **TikTec** [30]: A multimodal misinformation detection framework that leverages subtitles extracted from audio tracks and visual frames to effectively capture key information across modalities and improve detection performance.

- **FANVM** [9]: A topic-agnostic fake news video detection model based on adversarial learning and topic modeling. It dynamically adjusts the weight of comment features using stance estimation between titles/descriptions and comments through Gibbs sampling-based LDA.

- **SV-FEND** [24]: A multimodal approach for short video fake news detection that selects informative features via cross-modal correlation analysis and integrates social context using

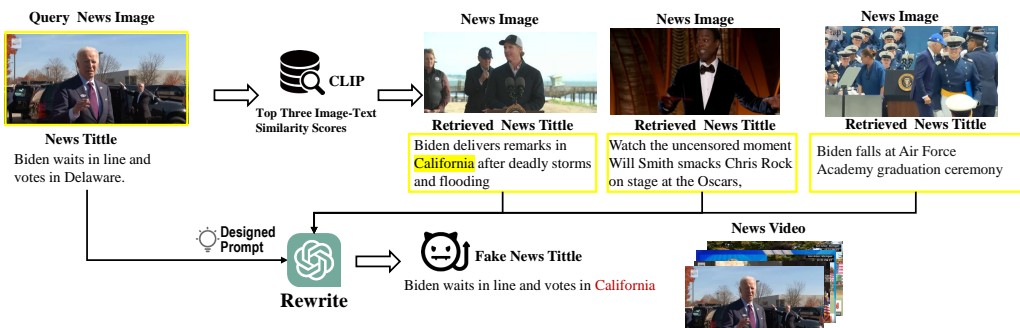

Figure 9: The candidate news sample retrieval pipeline based on Visual-to-Textual Retrieval matching employed in the FakeVV dataset.

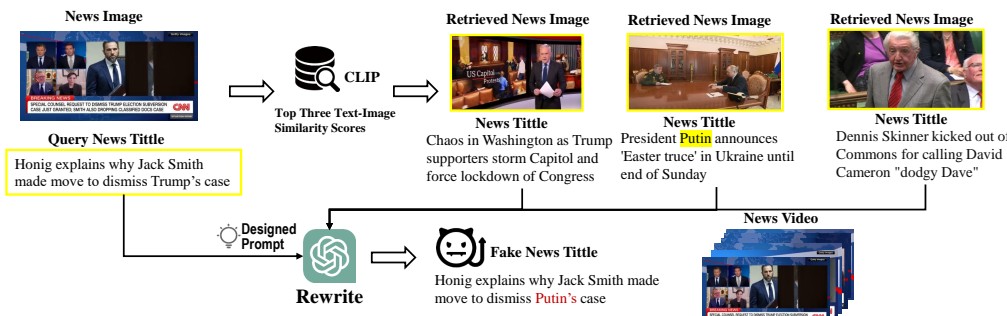

Figure 10: The candidate news sample retrieval pipeline based on Textual-to-Visual Retrieval matching employed in the FakeVV dataset.

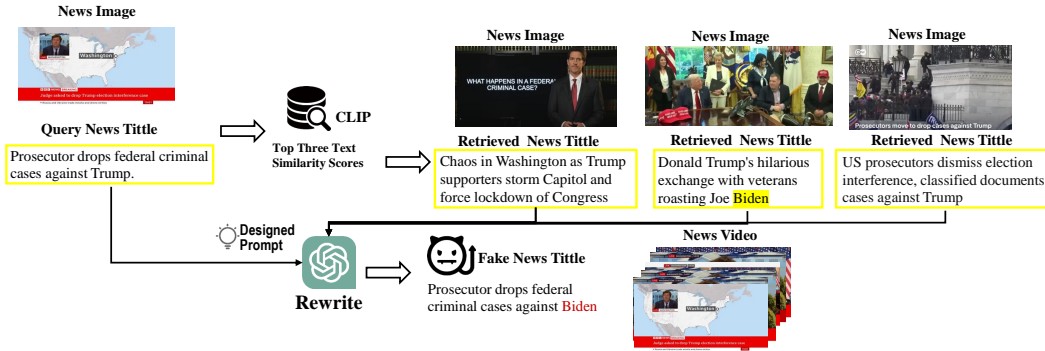

Figure 11: The candidate news sample retrieval pipeline based on Textual-to-Textual Retrieval matching employed in the FakeVV dataset.

co-attention and self-attention mechanisms. It captures inconsistencies between text, visual, and audio modalities to improve detection accuracy.

- **FakingRec** [6]: A short video misinformation detection method focusing on the "creative process" behind fake content. It employs a dual-branch network to model both material

selection (emotional and semantic preferences) and editing behavior (spatial and temporal features).

**(3) Closed-source MLLMs**:

- **GPT-4o** [1]: A state-of-the-art closed-source multimodal large language model from OpenAI, supporting visual and text inputs, known for its strong reasoning and generation capabilities.
- **Gemini2-thinking** [35]: A multimodal large language model developed by Google DeepMind, designed for advanced reasoning tasks across text and vision modalities.
- **GPT-o1-mini** [16]: A smaller-scale version of GPT-o1, optimized for reasoning and generation tasks with strong performance on text-based evaluations.
- **DeepSeek-R1** [14]: A powerful open-source multilingual large language model developed by DeepSeek, optimized for both natural language understanding and generation across diverse tasks.

**(4) Open-source MLLMs**:

- **Qwen2.5-VL** [3]: An open-source multimodal large language model supporting image and text inputs, with competitive performance on vision-language understanding benchmarks.
- **QVQ-72B-preview** [36]: A large-scale open-source vision-language model designed for visual reasoning tasks, preview version with 72 billion parameters.
- **InternVL2.5** [8]: A vision-language model optimized for cross-modal understanding, supporting fine-grained visual reasoning tasks.
- **InternVL2.5-78B-MPO** [39]: A scaled-up version of InternVL2.5 with 78 billion parameters, incorporating multimodal pre-training and optimization strategies for enhanced reasoning.

## A.3 Implementation Details

All experiments are conducted using PyTorch on $8\times$NVIDIA A100 GPUs, with Qwen2.5-VL as the base multimodal large language model (MLLM) and the default 8-frame sampling strategy for video inputs.

**Stage 1 (Long-CoT Tuning).** We apply LoRA for efficient fine-tuning and the LoRA hyperparameters are set as `lora_r=128` and `lora_alpha=256`, with the learning rate for the `mm_projector` set to $2 \times 10^{-6}$. The visual encoder remains frozen throughout this stage. LoRA fine-tuning is performed with a learning rate of $2 \times 10^{-5}$ and a batch size of 2. The model is trained for two epochs using the AdamW optimizer with a cosine annealing learning rate scheduler. This stage uses 85k high-quality instances generated from the Misinformation Long-CoT Generation process, derived from the FakeVV training set, with an equal split between fake and real samples.

**Stage 2 (DPO).** We train for one epoch on 5k human preference pairs from the FakeVV training set, using a batch size of 4 and a learning rate of $2 \times 10^{-5}$. The reference model is kept frozen, and gradient clipping is applied with a maximum norm of 1.0. We adopt the AdamW optimizer with a linear learning rate scheduler and warm-up over the first 500 steps. The DPO scaling factor $\beta$ is set to 0.1. Training is conducted with mixed precision (fp16) for improved efficiency.

**Stage 3 (GRPO).** We train for 172 steps using the verifiable reward function, sampling four candidate responses per input. The learning rate is set to $1\times10^{-6}$. Following the GRPO framework [32], rewards are computed based on reasoning correctness, entity-level alignment, and formatting consistency. Within each candidate group, rewards are normalized to estimate the relative quality of responses, which are then used to compute the importance-weighted advantages. Policy updates are performed with clipped advantage weighting and KL regularization against a frozen reference model to ensure training stability.

This stage utilizes 15k news samples for the main task, including training data from the FakeTT and FakeSV datasets. Additionally, we incorporate 3k *News Image OCR* and 3k *News Video Caption* samples as auxiliary tasks to enhance visual-textual reasoning. The rewards from auxiliary tasks are integrated with the main task rewards through a weighted sum to jointly guide the optimization.

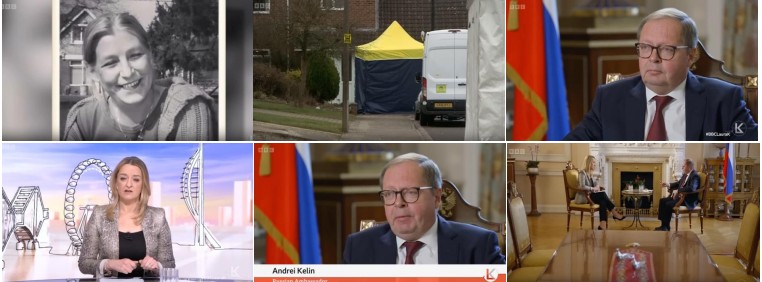

Russia's ambassador to UK appears to laugh and dismiss Nord Stream inquiry.

**(a) Event-type manipulation.**

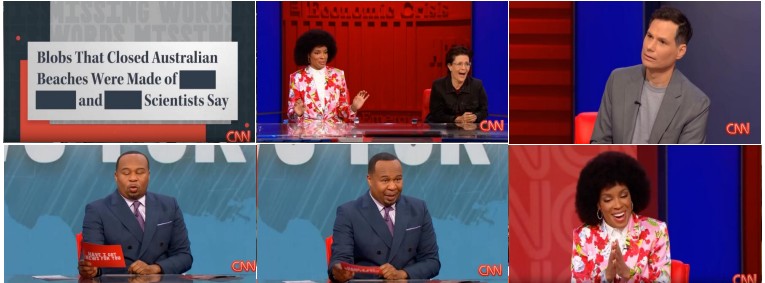

Kim Kardashian joins Roy Wood Jr. and his panel to play 'Lie-curious.

**(b) Person-type manipulation.**

Figure 12: Examples of Event-type and Person-type Manipulated Cases.

In Algorithm 1, we outline the GRPO training procedure. During training, the policy samples a group of responses for every input and assigns task-specific, verifiable rewards—accuracy-centred for OCR and captioning, and multi-component for misinformation detection (accuracy, format, key-word usage, and entity grounding). The rewards are z-normalized within each group to obtain advantages, which are then optimized with a clipped GRPO objective regularized by a KL term to the reference model.

## B  Analysis of News-domain Video Caption Generation

We introduce News-domain Video Caption as an enhanced description for news videos, which plays a critical role in both the Misinformation Long-CoT Generation and the GRPO with Verifiable Reward Function stages. Unlike conventional video captions, these captions incorporate specific news-related entities, enabling large language models to better understand multimodal video news.

We utilize the GPT-4o model to generate instruction-based, news-domain descriptions for video keyframes, guided by auxiliary information such as the original news title and associated entity names. Through carefully designed prompts in Figure 23, the generated captions are aligned with the factual content of each news image. Post-generation, we perform rigorous post-processing and validation to ensure the accuracy of the captions, confirming that the descriptions faithfully reflect the visual content and support effective learning of news-related semantics.

Figure 7 illustrates the data construction process and demonstrates how our approach produces accurate, fact-grounded video descriptions.

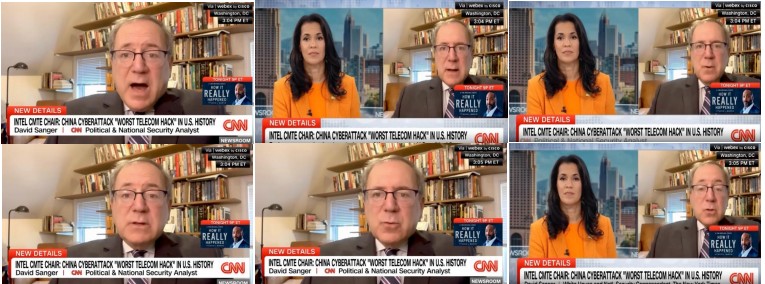

Officials call Russia's cyberattack 'worst telecom hack' in US history.

**(c) Location-type manipulation.**

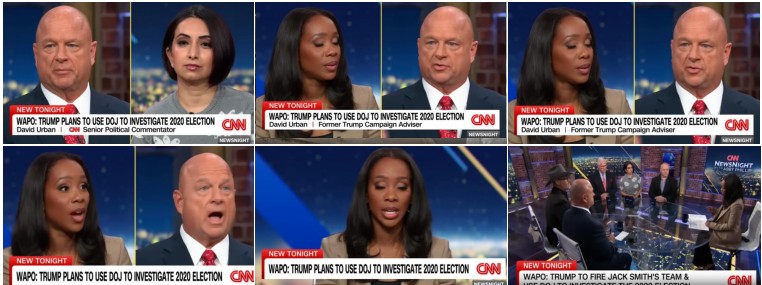

Trump is planning to use the FBI to address personal grievances.

**(d) Organization-type manipulation.**

Figure 13: Examples of Location-type and Organization-type Manipulated Cases.

## C  Analysis of FakeVV dataset

### C.1  Analysis of Misinformation Data Construction

To generate fine-grained manipulated news samples, we propose a challenging non-random entity replacement strategy to construct semantically inconsistent fake data. The overall workflow is described as follows: each news sample is represented as a pair (*Img*, *Title*), where *Img* denotes the first frame of the news video and *Title* refers to the corresponding news headline. We first extract visual and textual embeddings of each sample using the CLIP model to build a semantic news database. For any given query sample, we retrieve the top three most similar candidates from this database based on cosine similarity, providing entity references for subsequent manipulation.

The retrieval strategy randomly selects one of the following five methods:

- **Visual-to-Visual Retrieval**: As shown in Figure 8, the query image is used to retrieve the top three most similar images, along with their associated news titles. The query title "Three found alive and four bodies recovered after tourist boat capsizes in Red Sea" and the retrieved titles are then fed into GPT-4o with the designed prompt (Figure 25) to generate a fluent fake news title, such as "Three found alive and four bodies recovered after tourist boat capsizes in Mediterranean Sea."

- **Visual-to-Textual Retrieval**: As illustrated in Figure 9, the query image is used to retrieve the top three most similar text embeddings and their corresponding news titles. For example, given the query title "Biden waits in line and votes in Delaware," GPT-4o generates a fake title like "Biden waits in line and votes in California."

- **Textual-to-Visual Retrieval**: As illustrated in Figure 10, the query text (news title) is used to retrieve the top three most similar image embeddings and their associated titles. For example, the query title "Honig explains why Jack Smith made move to dismiss Trump's case" leads to the generated fake title "Honig explains why Jack Smith made move to dismiss Putin's case."

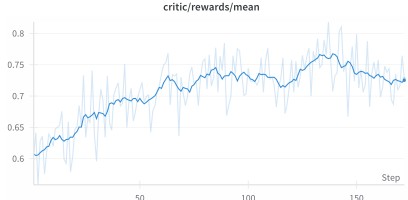

Figure 14: Reward Curve Over Training Steps.

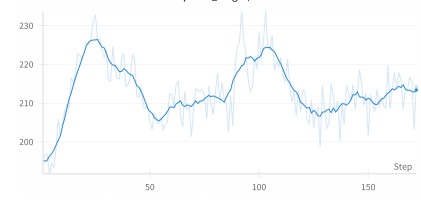

Figure 15: Response Length Curve Over Training Steps.

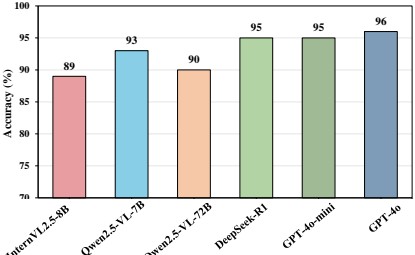

Figure 16: Consistency between MLLMs and human evaluation on explainability-correct samples from the FakeVV test set.

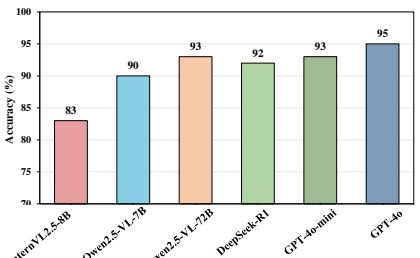

Figure 17: Consistency between MLLMs and human evaluation on explainability-error samples from the FakeVV test set.

- **Textual-to-Textual Retrieval**: As illustrated in Figure 11, the query title is used to retrieve the top three most similar textual samples. For instance, the query title "Prosecutor drops federal criminal cases against Trump" results in the fake title "Prosecutor drops federal criminal cases against Biden."

- **Random Selection**: Three random video-text pairs are selected from the database and used as reference inputs to GPT-4o for editing the query news title.

Based on the retrieved candidates, the query sample and the selected samples are input to GPT-4o, which replaces one randomly selected target entity in the query title using a carefully designed prompt, thereby generating manipulated news titles with semantic inconsistency.

## C.2 Additional Dataset Details

In this study, we show four types of manipulated news samples by replacing different categories of entities drawn from semantically similar samples: *Person*, *Location*, *Event*, and *Organization*. Figure 12 and Figure 13 present example cases for each manipulation type. Notably, the manipulated news titles exhibit minimal detectable artifacts and maintain fluent semantic coherence, making the detection task challenging. This design requires models to perform robust reasoning to accurately identify the inconsistencies.

## D  Prompt Analysis

**(1) Prompt for News-domain Video Caption Generation.** Figure 23 illustrates the prompt we designed for guiding GPT-4o to generate news-domain video captions. This prompt helps enrich visual information, enhancing the model's understanding of news content.

**(2) Prompt for Explainability Evaluation (Judge Model).** Figure 24 shows the prompt used to instruct GPT-4o to assess whether the model's reasoning process accurately identifies and describes the fake entity. This unified evaluation framework is used to obtain the interpretability metric for models on the FakeVV test set.

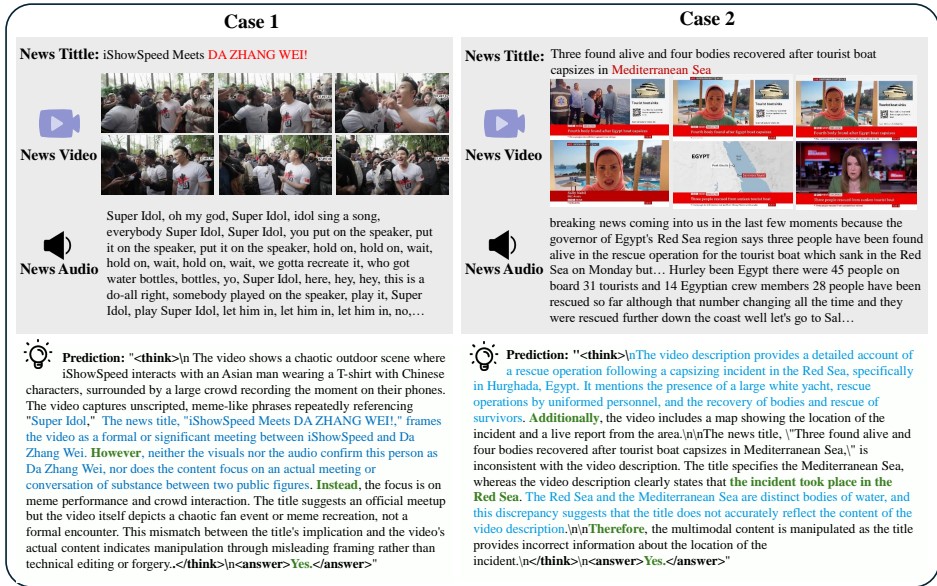

Figure 18: In two illustrative cases, Fact-R1 exhibits impressive deep reasoning abilities. Its long-form reasoning traces reveal not only a high degree of logical rigor and depth, but also precise identification of the manipulated (fake) entity.

**(3) Prompt for Misinformation Data Construction.** Figure 25 presents the prompt we use to guide GPT-4o in generating fluent fake news titles. The generation process leverages information from three reference titles to modify the original title, ensuring semantic plausibility while introducing manipulation.

# E    Additional Experiments

## E.1    GRPO Training Curves

Figure 14 and Figure 15 illustrate the reinforcement learning dynamics of Fact-R1 under the GRPO framework. During training, we monitor two key metrics: *Overall Reward* and *Response Length*. As shown in Figure 14, the overall reward exhibits a steady upward trajectory, indicating the model's continuous improvement in producing correct predictions throughout the reinforcement learning process. This trend suggests that our reward design effectively guides the model toward enhanced reasoning performance, thereby improving its capability in misinformation detection.

In addition, Figure 15 shows the dynamics of response length. Interestingly, we observe an initial decline in response length during the early stages of training, followed by a gradual increase and eventual stabilization around a consistent length. This behavior likely reflects an adaptive adjustment phase: the model first discards inefficient reasoning patterns inherited from supervised learning, then explores more concise responses during early reinforcement learning, and ultimately converges to a stable reasoning strategy that balances informativeness and efficiency. The convergence of response length indicates the emergence of a consistent reasoning policy optimized under the reward framework.

## E.2    Consistency Analysis Between Judge Model and Human Evaluation

To further assess the robustness of using GPT-4o-mini as the judge model in place of human evaluation for Fact-R1, we conduct a consistency analysis between several strong MLLMs and human judgments. Specifically, we manually select 100 samples from the FakeVV test set where Fact-R1's reasoning correctly identifies the labeled fake entity (explainability-correct samples) and another 100 samples where the reasoning fails to identify the correct entity (explainability-error samples).

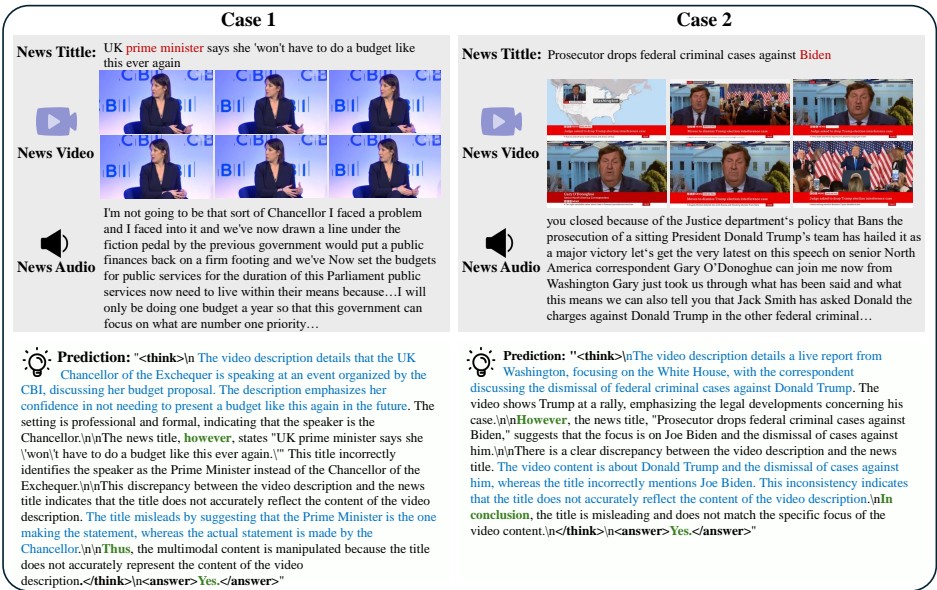

Figure 19: The figure presents two fake news cases successfully predicted by Fact-R1. In Case 1, the key information "Chancellor" from the news audio aids accurate detection, while in Case 2, the OCR-extracted text "Trump" from the news image provides critical visual evidence that supports the model's reasoning process.

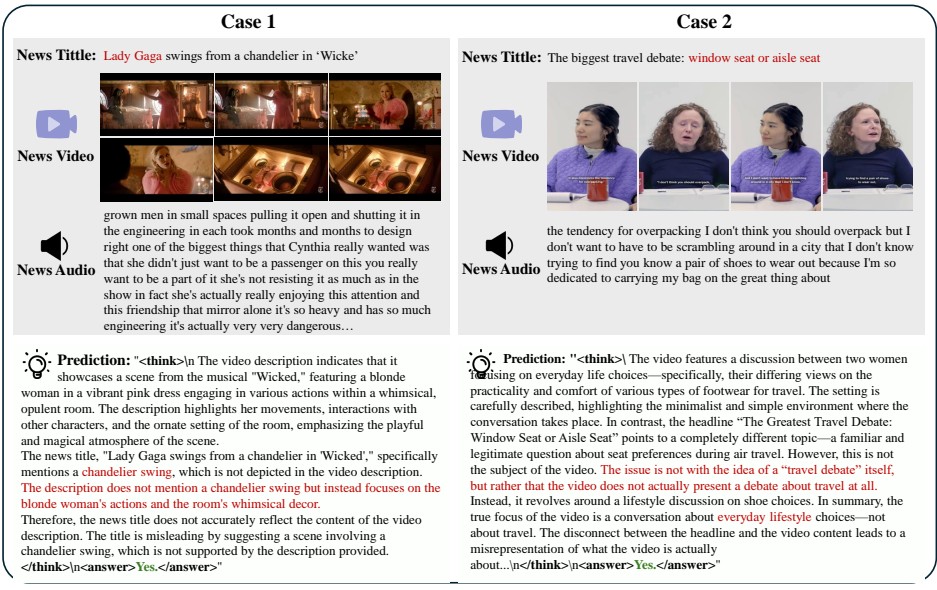

Figure 20: The figure presents two cases where Fact-R1 achieves correct label prediction but fails to accurately identify the fake entity in its reasoning process.

Using the same prompt design shown in Figure 24, we evaluate the consistency of five models—GPT-4o-mini, GPT-4o, DeepSeek-R1, Qwen2.5-VL-7B, and Qwen2.5-VL-72B—against human evaluations and report their accuracy on both types of samples.

As illustrated in Figure 16, for explainability-correct samples, DeepSeek-R1, GPT-4o-mini, and GPT-4o achieve high agreement with human evaluations, with negligible deviations. Figure 17 shows the results on explainability-error samples, where Qwen2.5-VL-72B, GPT-4o-mini, and GPT-4o again demonstrate high consistency with human assessments.

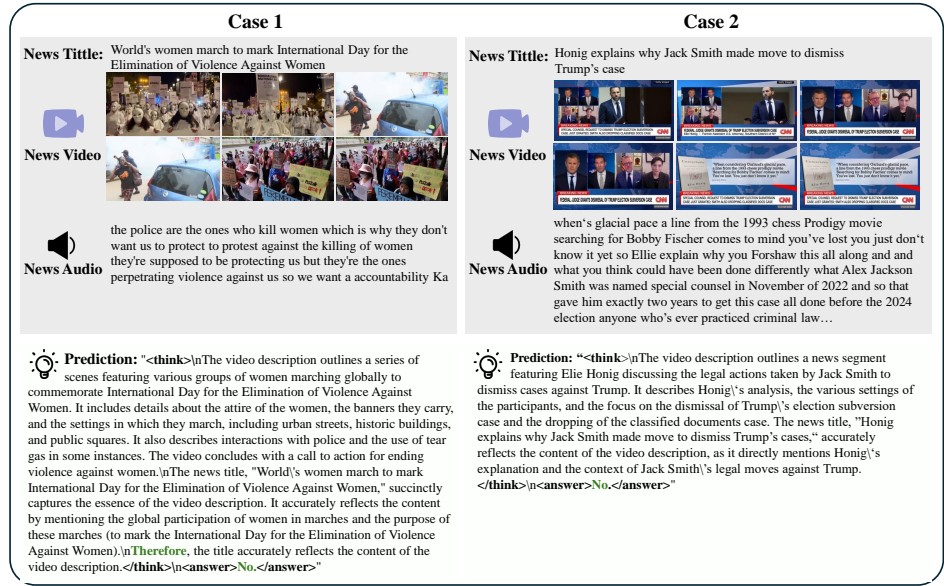

Figure 21: The figure presents two examples of real news cases successfully classified by Fact-R1.

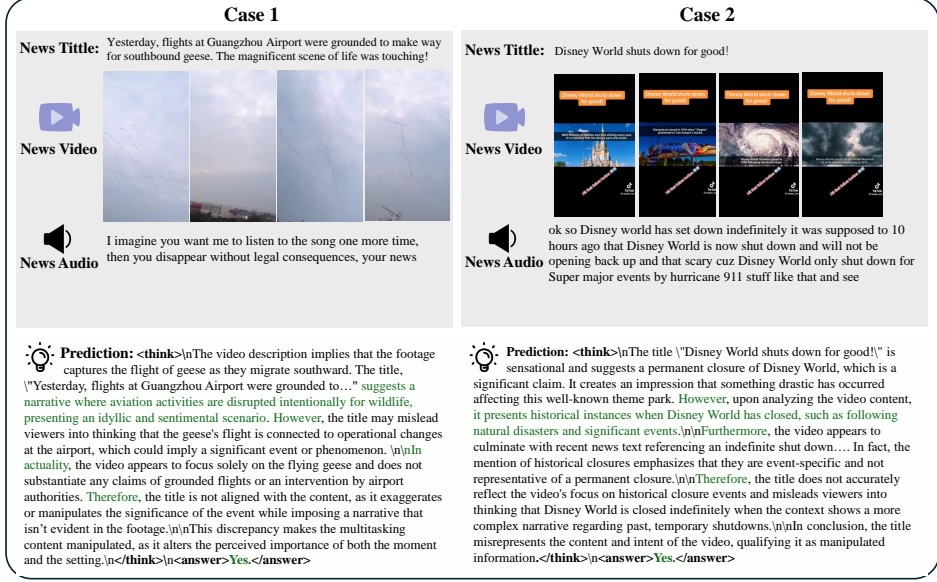

Figure 22: The figure presents two real-world fake news cases that were successfully predicted by Fact-R1.

Considering both cost-effectiveness and robustness, we adopt GPT-4o-mini as our judge model. Its evaluation accuracy remains within an acceptable error range, providing reliable and stable judgment for assessing reasoning quality in our misinformation detection framework.

### E.3 Case Study

**(1) Successful Reasoning and Entity Identification.** As shown in Figure 18, Fact-R1 demonstrates impressive deep reasoning capabilities, with key steps in its reasoning process highlighted in green. The model not only accurately determines whether a news video is manipulated but also generates structured and detailed reasoning chains that explicitly identify the fake entity. In Case 1, Fact-R1 infers from the visual content that the video depicts "Super Idol," which contradicts the title's

You are a video describer tasked with providing a concise and accurate description of a news video. Use the provided title and video image entity as contextual references to support your description.

# Title: {title}
# Entity: {entity}

# Instructions:
1. Briefly describe the main objects and their visual characteristics as shown in the video.
2. Summarize the environment and background setting.
3. Highlight key actions and interactions captured in the footage.

# Constraints:
1. Use a clear, narrative style. Avoid bullet points or lists.
2. Keep the description brief and focused on the most essential details.
3. Avoid speculation, subjective opinions, or emotional language.
4. The provided entity may contain inaccuracies and should be treated only as a reference.

Figure 23: Prompt design for generating news-domain video captions.

reference to "DA ZHANG WEI." In Case 2, by extracting and interpreting key details from the news report, the model deduces that the event occurred in the Red Sea, successfully identifying the inconsistency with the location mentioned in the title and confirming the video as manipulated.

**(2) Contribution of Multimodal Signals.** As shown in Figure 19, we present two fake news cases correctly identified by Fact-R1, demonstrating the importance of multimodal evidence in the reasoning process. In Case 1, the presence of the key term "Chancellor" in the news audio provides critical information that aids accurate detection. In Case 2, the OCR-extracted text "Trump" from the news image serves as essential visual evidence, effectively supporting the model's reasoning and final judgment. These examples highlight Fact-R1's ability to integrate cross-modal information for robust misinformation detection.

**(3) Failure Cases in Explainability.** Figure 20 illustrates two cases where Fact-R1 successfully predicts the correct label but fails in its explainability by inaccurately identifying the fake entity. In Case 1, the ground-truth fake entity is "Lady Gaga," yet the model mistakenly identifies "chandelier" as the manipulated entity. In Case 2, the correct fake entity is "window seat or aisle seat," but the model incorrectly selects "travel debate" as the fake entity. Upon analysis, we observe that in both cases, neither the news audio nor the OCR-extracted text from the news images provides effective supporting clues. This indicates a limitation in the model's reasoning ability when explicit cross-modal signals are absent, suggesting the need for enhanced entity grounding mechanisms and deeper semantic understanding.

**(4) Accurate Reasoning on Real News Samples.** As shown in Figure 21, we present two examples of real news videos correctly identified by Fact-R1. In these cases, the model's reasoning process accurately describes the events occurring in the video and aligns well with the given news titles. These examples demonstrate Fact-R1's capability not only in detecting manipulated content but also in validating authentic news through consistent and coherent reasoning.

**(5) Real-world Cases.** As shown in Figure 22, we present two real-world fake news cases that were successfully detected by Fact-R1. These cases are drawn from the *FakeSV* and *FakeTT* datasets, which

You are an AI assistant to help me evaluate whether the reasoning process in the prediction correctly identifies the labeled fake entity.

Task Description:

Determine if the following prediction mentions the fake entity: "{**entity**}".

Prediction: "{**prediction**}"

Output "yes" if the prediction reasoning process mentions and describes the entity; otherwise, output "no".

Example Case:
Fake Entity: "French"
Prediction: "The news title, 'Storm Bert: drone footage shows extent of flooding at French holiday park,' suggests that the flooding occurred at a French holiday park, which is inconsistent with the video description indicating that the flooding occurred at an English holiday park. Thus, the news title does not accurately reflect the content of the video description, as it incorrectly states the location of the flooding. In conclusion, the event discussed in the news title does not match the specific location of the flooding depicted in the video description."
Output: yes

Figure 24: Prompt design for the judge model to evaluate the accuracy of fake entity identification in the model's reasoning process.

consist of fabricated information posted by users. We observe that Fact-R1 is able to successfully capture the key indicators of misinformation in these examples.

## F   Responsible Release and Safeguards

To mitigate potential misuse risks associated with misinformation detection models, the FakeVV dataset is provided strictly for non-commercial research purposes and is accessible only to verified academic researchers under a research-specific license agreement. All samples have been carefully screened to remove any non-public personally identifiable information (PII), ensuring that no harmful, private, or illegal content is included. Fact-R1 is intended to function as an assistive component within human-in-the-loop workflows, rather than as a standalone decision-making or content moderation system. These measures are designed to balance research transparency with the responsible release of models and data.

## G   Ethics Statement

This work includes human evaluation via crowdsourced annotation. Annotators received clear task instructions and example interfaces, provided informed consent, and could withdraw at any time. Responses were anonymized, no personally identifiable information was collected, and compensation met or exceeded local fair-pay standards. The study adhered to institutional ethical guidelines and local labor regulations.

We collected a small annotation set (5k preference samples) under a transparent, compliant protocol. Given the non-sensitive content and minimal risk typical of standard annotation tasks, the study quali-

You are an AI assistant. Your task is to generate a fake news title by modifying the given original title, using the three provided unrelated reference titles as inspiration.

# Task Description:
You will be provided with:
1. An original news title.
2. Three unrelated reference news titles.

Your goal is to create a **fake news title** by replacing key information in the original title. The replacement should be inspired by the entities or context found in the reference titles.

# Instructions:
- Replace specific details in the original title, which may include:
  - **Person Information:**
    Example:
    From "Macron corrects Donald Trump on costs of Ukraine war and says: 'Peace must not mean surrender'"
    To "Angela Merkel corrects Donald Trump on costs of Ukraine war and says: 'Peace must not mean surrender'."
  - **Location Information:**
    Example:
    From "Volodymyr Zelenskiy calls for air defence boost during Antony Blinken visit to Ukraine"
    To "Volodymyr Zelenskiy calls for air defence boost during Antony Blinken visit to Russia."
  - **Event Information:**
    Example:
    From "Myanmar fitness coach accidentally captures a military coup"
    To "Myanmar fitness coach accidentally captures a peaceful protest."
  - **Organization Information:**
    Example:
    From "Apple pulls data protection tool after UK government security row"
    To "Xiaomi pulls data protection tool after UK government security row."

- After generating the fake title, identify **which category the modification belongs to** (choose from: person, location, event, organization).

# Example:
{{
  "title": "Three found alive and four bodies recovered after tourist boat capsizes in Red Sea.",
  "reference_title1": "That Time the Mediterranean Sea Disappeared.",
  "reference_title2": "Sea levels rose more than expected in 2024.",
  "reference_title3": "Antarctica's Weddell Sea 'deserves protected status.'",
  "Generate Fake Title": "Three found alive and four bodies recovered after tourist boat capsizes in Mediterranean Sea.",
  "Alteration Category": "location"
}}

# Now, process the following input:
Original Title: "{title}"
Reference Title 1: "{Reference _title1}"
Reference Title 2: "{Reference _title2}"
Reference Title 3: "{Reference _title3}"

Please output the result in the following JSON format:
{{
  "Generate Fake Title": "...",
  "Alteration Category": "..."
}}

Figure 25: Prompt design for generating fluent fake news titles by editing the original title based on information from three reference titles.

fied for an IRB exemption under our institution's policy. We support transparency and reproducibility and have shared data and experimental details responsibly, prioritizing ethical considerations, participant privacy, and societal safety.

