# OpenReview forum: "Fact-R1: Towards Explainable Video Misinformation Detection with Deep Reasoning"
_NeurIPS.cc/2025/Conference — NeurIPS 2025 poster_

### Official Review · Reviewer_riKt · 2025-06-28

**Clarity:** 3
**Significance:** 4
**Originality:** 3
**Rating:** 5
**Confidence:** 5

**Summary:**

The paper tackles an important problem, video misinformation detection. The paper argues that the most common and misleading types of misinformation involve using real videos out of context that is taking genuine footage and placing it within a false or distorted narrative to deceive viewers. To build/train models to detect such content, there is a lack of datasets especially for multi-modal large language models (MLLMs). Hence the paper proposes a multi-modal dataset with 100,000 video-text pairs.

The authors collected 100,000 video-text pairs from four news outlets, BBC News, Guardian News, CNN, and The New York Times, from November 2006 to February 2025. Along with the videos, they preserved metadata like user comments, likes, and timestamps. They used a GPT-4o to generate detailed, news-specific captions using video titles and named entities.

To create misinformation examples, they introduced a method that swaps key entities (like people, places, events, and organizations) with semantically inconsistent alternatives. This was done using CLIP-based similarity matching to find closely related news content, making the fake samples appear plausible. The dataset was also refined by removing non-news videos, filtering out long content, and eliminating near-duplicates through clustering of video titles.

Then they train the Qwen2.5-VL model in three stages, tuning LORA layers using long-chain-of-thought (long-CoT) instruction tuning (85k samples), DPO tuning on 5k samples, and then GRPO training with Accuracy formate, reasoning, and entity reward while making the reward verifiable by which the authors mean that reasoning traces are only useful if model made a correction decision about the video being fake or real.

**Questions:**

What is the input resolution of evaluation videos? What is the training input resolution for your model? Are all the models in Table 2 trained and tested on the same resolutions?

What is the generalization trade-off of the Qwen2.5-VL model? How much does Qwen2.5-VL lose in terms of generalization after tuning on the proposed dataset?

**Ethical Concerns:**

["NO or VERY MINOR ethics concerns only"]

**Final Justification:**

My main concerns, especially regarding generalization, have been addressed by the rebuttal. Considering this, I have improved my rating.

**Limitations:**

Yes

**Paper Formatting Concerns:**

Did not notice formatting issues.

**Quality:**

3

**Strengths And Weaknesses:**

The tackled problem is significant. The paper is well-written and organized. The generated data pipeline is clearly described and reproducible, and so are the training details. The results are favorably better than baseline Qwen 2.5-VL and similar approaches on three benchmarks/datasets (FakeSV, FakeTT, FakeVV).

The originality of this work mainly lies in creating a large-scale dataset and fine-grained reward modifications by invoking semantic consistency  (that is, reasoning matters when the model is accurate) during GRPO training. The paper is not technically novel as it uses existing approaches to train the model on the proposed dataset.

---

> ### Author Rebuttal · Authors · 2025-07-31
>
> Thank you very much for your thoughtful and constructive feedback. Your insightful comments have been extremely valuable in helping us improve our work, and we have carefully addressed them in the revised version. Below, we restate each of your comments followed by our point by point responses. We are deeply grateful for the valuable time and thoughtful effort you devoted to reviewing our paper.
>
> >**Q1:** What is the input resolution of evaluation videos? What is the training input resolution for your model?
>
> **A:** Thank you for your question. Unlike the approach of forcing all inputs to a fixed resolution (e.g., 224×224), Fact‑R1, like Qwen2.5‑VL, uses a dynamic resolution handling mechanism. This allows us to process video frames at their original resolution and map them into visual patch tokens without the need for forced resizing. This design helps retain the spatial proportions and contextual information better during encoding.
> - For each video, we sample key frames as input, following Qwen‑VL's default video processing strategy.
> - The sampled frames are input at their original resolution and are dynamically mapped into patch tokens by Qwen2.5‑VL.
> - We do not resize the frames to 224×224, thus maintaining the strategy consistent with the pre-training configuration of Qwen2.5‑VL.This approach avoids the distortion caused by fixed resizing and aligns with the pre-training configuration of Qwen2.5‑VL.
>
> To summarize, Fact‑R1 uses original resolution for input during both training and evaluation, and the dynamic resolution handling ensures consistency with Qwen 2.5‑VL's pre-training setup.
>
> >**Q2:** Are all the models in Table 2 trained and tested on the same resolutions?
>
> **A:** (1) We sincerely thank the reviewer for their valuable suggestions. Except for the closed-source MLLMs and open-source MLLMs, we performed fine-tuning on all other baselines using the same training set, which includes portions of FakeSV, FakeTT, and FakeVV. The input resolution for both the training and testing datasets remains consistent with the original news video. During the training and testing process, we followed the appropriate visual encoder and cropping strategies for each baseline to ensure optimal performance.
>
> (2) Thank you for your insightful comments. We have added fine-tuning experiments with open-source MLLMs, including Qwen2.5-VL-7B and InternVL2.5-8B, using label data for the answers. Due to practical application scenarios and computational resource constraints, we have not yet conducted full fine-tuning on large-scale models.
>
> | Dataset | Method              | Acc   | F1    |
> | :-----: | :-----------------: | :---: | :---: |
> | FakeSV | Qwen2.5-VL-7B-SFT              | 71.8  | 72.1  |
> | | InternVL2.5-8B-SFT         | 70.1  | 66.1  |
> |  | Fact-R1  | 75.6  | 74.7  |
> | FakeTT  | Qwen2.5-VL-7B-SFT              | 68.8  | 66.4  |
> |   | InternVL2.5-8B-SFT         | 67.7  | 68.0  |
> |  | Fact-R1  | 74.4  | 72.7  |
> | FakeVV | Qwen2.5-VL-7B-SFT              | 75.4  | 73.3  |
> |  | InternVL2.5-8B-SFT         | 73.5  | 73.2  |
> |  | Fact-R1  | 81.2  | 80.3  |
>
> As shown in the table, fine-tuning Qwen2.5-VL-7B achieved impressive performance, further validating its suitability as the backbone for Fact‑R1. The best performance of Fact‑R1 demonstrates the importance of incorporating deep reasoning in misinformation detection.
>
> >**Q3:** What is the generalization trade-off of the Qwen2.5-VL model?
>
> **A:** (1) Thank you for your question. The strong performance of Fact‑R1 is largely attributed to Qwen2.5‑VL, a model that introduces extensive reasoning-focused data during its pre-training phase, making it highly suitable as the backbone for deep reasoning tasks. Several studies have shown that Qwen-based models excel in multi-step reasoning and complex multimodal understanding tasks, which further validates the choice of Qwen2.5‑VL as the backbone for our work.
>
> (2) To analyze the trade-offs in its generalization ability, we conducted ablation experiments using different open-source Multimodal Large Language Models, including InternVL2.5‑8B and LLaVA‑Video‑7B [9].
>
> | Dataset | Method              | Acc   | F1    |
> | :-----: | :-----------------: | :---: | :---: |
> | FakeSV | LLaVA-Video-7B              | 72.4  | 70.9  |
> | | InternVL2.5-8B         | 72.1  | 71.7  |
> | | Fact-R1  | 75.6  | 74.7  |
> | FakeTT  | LLaVA-Video-7B              | 69.0  | 68.3  |
> |  | InternVL2.5-8B         | 72.0  | 71.8  |
> |  | Fact-R1  | 74.4  | 72.7  |
> | FakeVV | LLaVA-Video-7B              | 74.9  | 73.8  |
> | | InternVL2.5-8B         | 77.7  | 76.4  |
> |  | Fact-R1  | 81.2  | 80.3  |
>
>
> As shown in the table, the results indicate that when Qwen2.5‑VL is used as the backbone, Fact‑R1 achieves the best performance in terms of prediction accuracy, demonstrating the model's excellent chain-of-thought reasoning quality and factual generalization ability.
>
> >**Q4:** How much does Qwen2.5-VL lose in terms of generalization after tuning on the proposed dataset?
>
> **A:** We sincerely thank the reviewer for their valuable feedback. We evaluated the generalization performance of Fact‑R1 after fine-tuning on our dataset using several widely adopted open-source benchmarks.
> - OCRBench [1] evaluates optical character recognition and visual-text understanding.
> - HallusionBench [2] measures model hallucination and factual consistency in multi-modal reasoning.
> - MMMU_VAL [3], MMStar [4], and MMVet [5] are multimodal evaluation benchmarks covering reasoning, commonsense, and world knowledge.
> - AI2D [6] and MathVista [7] assess diagram and math-related reasoning.
> - MME [8] is a comprehensive multi-modal evaluation benchmark across multiple tasks.
>
> All evaluations follow the OpenCompass framework to ensure fairness and reproducibility.
>
> | Dataset| Method| Score  |
> | :---: | :--: | :--: |
> | OCRBench | Qwen2.5-VL-7B   | 888    |
> |   | Fact-R1|875|
> | HallusionBench| Qwen2.5-VL-7B   | 51.9   |
> | | Fact-R1| 48.8|
> | MMMU_VAL| Qwen2.5-VL-7B| 58|
> | | Fact-R1| 54|
> | MMStar        | Qwen2.5-VL-7B   | 64.1   |
> |               | Fact-R1         | 62.6   |
> | MMVet         | Qwen2.5-VL-7B   | 69.7   |
> |               | Fact-R1         | 62.9  |
> | AI2D          | Qwen2.5-VL-7B   | 84.3   |
> |               | Fact-R1         | 83.0   |
> | Mathvista     | Qwen2.5-VL-7B   | 68.1   |
> |               | Fact-R1         | 62.9   |
> | MME           | Qwen2.5-VL-7B   | 2312.1 |
> |               | Fact-R1         | 2209.3 |
>
> As shown in the table, Fact-R1 retains most of Qwen2.5-VL's original capabilities, with only a small drop in overall performance due to the lightweight nature of our training pipeline.
> Notably, Fact-R1 performs well on HallusionBench and OCRBench, which is consistent with the additional reasoning-related and OCR-related data introduced in the DPO and GRPO stages. This indicates that our fine-tuning successfully preserved capabilities closely related to misinformation reasoning, while sacrificing performance on tasks like MathVista and MMVet, which are less relevant to our target application.
> We will include a more comprehensive analysis of these results and release the evaluation code in the final supplementary material for community verification.
>
> >**Q5:** The paper is not technically novel as it uses existing approaches to train the model on the proposed dataset.
>
> **A:** Thank you for recognizing the Fact‑R1 pipeline and providing a clear summary of our work. Fact‑R1 is not a simple application of GRPO to a classification task, but rather a systematic exploration of video misinformation detection, incorporating several key innovations:
>
> - **Task‑specific verifiable rewards.** Unlike [10], which adopts uniform classification rewards, Fact‑R1 introduces entity recognition rewards ($R_{\text{entity}}$) and reasoning keyword rewards ($R_{\text{word}}$). A judge model is employed to simulate human auditors, guiding the model to produce reasoning chains that are both accurate and interpretable.
>
> - **Auxiliary tasks for multimodal enhancement.** Motivated by failure case analysis and inspired by the reasoning process in Figure 20, we first incorporate News Image OCR and News Video Caption generation as auxiliary tasks, designing dedicated rewards for each. These tasks strengthen the model’s ability to understand multimodal information and handle entity‑level reasoning in manipulated content.
>
> - **Progressive reinforcement learning framework.** The three‑stage pipeline (Long‑CoT → DPO → GRPO) is carefully designed to combine explainable data construction, preference alignment, and task‑specific reward optimization. Each stage progressively improves reasoning quality, factual consistency, and interpretability.
>
> Overall, Fact‑R1 contributes a large‑scale, finely annotated dataset (FakeVV) together with a task‑specialized reinforcement learning framework. These innovations enable Fact‑R1 to achieve strong predictive performance while generating verifiable, human‑aligned reasoning chains, advancing practical and explainable video misinformation detection.
>
> [1] Liu et al. Ocrbench: on the hidden mystery of ocr in large multimodal models.
>
> [2] Guan et al. Hallusionbench: an advanced diagnostic suite for entangled language hallucination and visual illusion in large vision-language models.
>
> [3] Yue et al. Mmmu: A massive multi-discipline multimodal understanding and reasoning benchmark for expert agi.
>
> [4] Chen et al. Are we on the right way for evaluating large vision-language models?
>
> [5] Yu et al. Mm-vet: Evaluating large multimodal models for integrated capabilities.
>
> [6] Kembhavi et al. A diagram is worth a dozen images.
>
> [7] Lu et al. Mathvista: Evaluating mathematical reasoning of foundation models in visual contexts.
>
> [8] Fu et al. Mme-survey: A comprehensive survey on evaluation of multimodal llms.
>
> [9] Canhui Tang. Open llava-video-r1. Open-LLaVA-Video-R1, 2025.
>
> [10] Liu et al. Visual-rft: Visual reinforcement fine-tuning.

---

> > ### Comment · Reviewer_riKt · 2025-08-09
> > **Thanks for detailed rebuttal**
> >
> > I thank the authors for the detailed rebuttal, which addressed my concerns. I will consider this in the final rating.

---

> > > ### Author Response · Authors · 2025-08-09
> > >
> > > We are truly grateful for your constructive feedback and for the time you kindly dedicated to reviewing our response once again. Your engagement is highly valued, and we will incorporate your suggestions into the final version of the manuscript.

---

### Official Review · Reviewer_19bT · 2025-07-02

**Clarity:** 3
**Significance:** 3
**Originality:** 3
**Rating:** 5
**Confidence:** 4

**Summary:**

This paper introduces FakeVV and Fact-R1, a large-scale dataset of 100,000 video-text pairs designed to advance multimodal misinformation detection with a novel framework combining deep reasoning and collaborative reinforcement learning. FakeVV features realistic misinformation samples created through semantically consistent entity replacements, supporting interpretability of the method. Experimental results confirm that Fact-R1 significantly surpasses existing methods in detection accuracy and explainability.

**Questions:**

1. How well would Fact-R1 generalize to non-news domains?

2. Can you provide detailed computational analysis or complexity metrics for each component?

3. Can you give comparisons with recent methods using variants of GRPO? DeepVideo-R1 https://www.arxiv.org/abs/2506.07464, Video-R1 https://arxiv.org/abs/2503.21776 etc..

**Ethical Concerns:**

["NO or VERY MINOR ethics concerns only"]

**Final Justification:**

The additional evaluations in the medical domain and multiple benchmarks strengthen the generalization claim, and the computational breakdown addresses efficiency and reproducibility concerns. Comparisons with GRPO variants are convincing, I will reflect this and keep my final score as accept.

**Limitations:**

yes

**Paper Formatting Concerns:**

No concerns

**Quality:**

3

**Strengths And Weaknesses:**

Strengths :

- FakeVV is a large-scale dataset covering a broad range of topics and time frames, significantly advancing the field. And integration of deep reasoning with reinforcement learning (Long-CoT, DPO, GRPO) is effectively addresses critical limitations of existing methods. They're technically sounds.
- Fine-grained annotation enables interpretability of this method, which is crucial for practical misinformation detection.
- Extensive evaluations clearly show Fact-R1’s state-of-the-art performance compared to baselines
- They validate each methodological component through ablation studies.
- Extensive supplementary materials and scripts for reproducing their results.
- Writing is clear and easy to follow.


Weaknesses :

- The proposed training approach, while helpful, may require significant computational resources and extensive training time. There should be some computational analysis too.
- Heavy reliance on external models for caption generation make concerns on reproducibility and independence.
- Need further evaluation on even broader real-world datasets and scenarios beyond short news clips might be necessary to ensure generalizability.
- The dataset primarily focuses on political and news-related topics, limiting applicability to other misinformation types (scientific or healthcare misinformation)

---

> ### Author Rebuttal · Authors · 2025-07-31
>
> Thank you for your constructive comments and suggestions, and they are exceedingly helpful for us to improve our paper. We have carefully incorporated them in the revised paper. In the following, your comments are first stated and then followed by our point-by-point responses.
>
> >**Q1:** How well would Fact-R1 generalize to non-news domains? Need further evaluation on even broader real-world datasets and scenarios beyond short news clips might be necessary to ensure generalizability.
>
> **A:** We sincerely thank the reviewer for their valuable comments. We have evaluated the generalization capability of Fact-R1 in non-news domains from the following two aspects:
>
> (1) **Performance on Non-News Misinformation Datasets.** We tested Fact-R1's applicability in the medical domain on the Covid-VTS [1] dataset. Each video in Covid-VTS contains a long real or fake claim. Following the experimental setup of TwtrDetective [1], we used video frames, OCR text, speech text, and claim text as inputs for binary misinformation detection.
>
> |Dataset|Method|Acc|F1|
> |:-:|:-:|:-:|:-:|
> | COVID-VTS |GPT-4o| 55.9 |55.7|
> ||TwtrDetective|68.1|67.9|
> ||Fact-R1|71.6|69.8|
>
> The results show that Fact-R1 performs at a level close to task-specific baseline models for medical-related video misinformation. In the future, we plan to enrich FakeVV with additional medical domain data to better support training and evaluation of open-source models.
>
> (2) **Preservation of Generalization Capability.** We further evaluated Fact-R1's generalization capability on several open-source benchmarks.
> |Dataset | Method| Score|
> |:-:|:--:|:--:|
> |OCRBench|Qwen2.5-VL-7B|888|
> | | Fact-R1|875|
> | HallusionBench| Qwen2.5-VL-7B| 51.9   |
> | |Fact-R1| 48.8|
> | Mathvista| Qwen2.5-VL-7B| 68.1|
> | | Fact-R1| 62.9|
>
> All evaluations follow the OpenCompass framework to ensure fairness and reproducibility. As shown in the table, Qwen2.5-VL-7B reflects the pre trained model’s baseline performance. Thanks to the introduction of the DPO phase and the enhancement of GRPO auxiliary tasks, Fact-R1 performed well on HallusionBench [6] and OCRBench [5], tasks that are highly relevant to reasoning and multimodal misinformation understanding. However, we observed a noticeable decline in performance on MathVista[7], indicating a reduction in its mathematical reasoning capability.
>
> We will include these results in the final version of the manuscript and make the evaluation code publicly available for the community to reproduce and validate.
>
> >**Q2:** Can you provide detailed computational analysis or complexity metrics for each component? The proposed training approach, while helpful, may require significant computational resources and extensive training time. There should be some computational analysis too.
>
> **A:** Thank you for your suggestion. In designing Fact-R1, we carefully considered both practical efficiency and reproducibility. We used a 7B parameter backbone model and avoided expensive pre-training. All experiments were conducted using PyTorch on multiple 8 × A100 GPUs. Our framework is lightweight and fully transparent. The computational cost for each stage is summarized as follows:
>
> | Training Stage| Dataset Size|Trainable Params|BatchSize|Time|GPU|
> |-|-|-|-|-|-|
> | Stage 1 (Long CoT Tuning) |85k instances|LoRA|2| ~22 h| 8 × A100|
> | Stage 2 (DPO)| 5k preference pairs|LoRA| 1 | ~1 h| 8 × A100|
> | Stage 3 (GRPO)| 15k main + 5k auxiliary | Full model fine-tuning |1| ~9 h| 8 × A100|
>
> Overall, the total training time is under 35 hours, and the framework does not require any large-scale pre-training, making it both highly efficient and easy to reproduce, which is convenient for community researchers.
> We will include this computational analysis table in the final version to enhance transparency.
>
>
> >**Q3:** Can you give comparisons with recent methods using variants of GRPO?
>
> **A:** We sincerely thank the reviewer for their valuable suggestions. Recently, several impressive works have explored variants of GRPO:
> - DeepVideo-R1 [2] redefined the traditional GRPO reinforcement learning objective as a regression task, directly predicting the advantage values of a sample set, and dynamically adjusted the difficulty of the video-text input by incorporating reasoning hints or altering the video content. We highly appreciate its contributions in long video reasoning and plan to conduct detailed comparative experiments once its algorithm is open-sourced.
> - Video-R1 [3] enhanced temporal modeling capabilities and designed a reward function with timeline clue enhancement to improve the model's understanding and reasoning of dynamic visual content, providing valuable insights for video reasoning.
> - DAPO [4] is an improved reinforcement learning method that decouples clipping, dynamic sampling, token-level loss, and length-aware reward shaping, significantly improving large model stability and accuracy in complex chain-of-thought reasoning tasks.
>
> To ensure fair comparison, We conducted the same reinforcement training on the third-stage data for Video-R1 to assess its performance on our task. At the same time, we replaced the third-stage GRPO algorithm in the Fact-R1 pipeline with DAPO to directly compare the differences between the two optimization methods.
>
> | Dataset | Method              | Acc   | F1    |
> | :-----: | :-----------------: | :---: | :---: |
> | FakeSV | Video-R1  | 69.9  | 68.4  |
> |  | DAPO  | 72.4  | 72.1  |
> |  | Fact-R1  | 75.6  | 74.7  |
> | FakeVV | Video-R1  | 74.3  | 72.7  |
> |  | DAPO  | 79.1  | 78.8  |
> |  | Fact-R1  | 81.2  | 80.3  |
>
>
> The experimental results show that Fact-R1 still maintains a leading position across all metrics. We attribute this to our task-specific reward functions designed for misinformation detection, as well as the novel reinforcement learning framework that integrates multiple auxiliary tasks. For example, the timeline clue enhancement reward in Video-R1 mainly improves the model's fine-grained understanding of temporal information but provides limited benefits to the misinformation detection task, while our task-specific GRPO design significantly enhances the model's reasoning ability in video misinformation detection.
> We will include these experimental results and relevant references in the revised manuscript.
>
>
>
> >**Q4:** Heavy reliance on external models for caption generation make concerns on reproducibility and independence.
>
> **A:** We sincerely thank the reviewer for raising this important question.
> - Reproducibility. Our news video captioning process is fully transparent. In *Section Data Pre-processing (L130)*, we provide a detailed description of the process for generating News-domain Video Captions and showcase the specific prompts used with GPT-4o in Figure 24. Additionally, we have already made part of the generated training data available in the supplementary materials, and we will fully release all video captions after the review period to ensure the reproducibility of our results.
> - Independence. We understand the reviewer’s concern regarding the reliance on closed-source models. In fact, the process of generating News-domain Video Captions in FakeVV is model-agnostic, meaning that any open-source captioning model can be used to replace GPT-4o without affecting the subsequent training framework. With the full release of FakeVV, researchers can either directly use the publicly available caption data or regenerate captions using any model, ensuring both the dataset and methodology remain independent.
>
>
>
> >**Q5:** The dataset primarily focuses on political and news-related topics, limiting applicability to other misinformation types (scientific or healthcare misinformation)
>
> **A:** (1) We sincerely thank the reviewer for their valuable feedback. Currently, the FakeVV dataset contains over 100k high-quality video–text pairs from four official news channels, aimed at advancing the development of community technology. We plan to expand the dataset by including more diverse news outlets, such as Fox News, ABC News, and others, to further enhance the variety of data sources.
>
> (2) Our preliminary analysis of the news topics in FakeVV shows that health and science account for only 13.6% and 7.9%, respectively, while the majority of samples focus on social, political, and cultural issues. To address this imbalance, we are expanding the dataset with additional science- and healthcare-related video–text pairs from specific news categories. The data collection pipeline is easily adaptable to various domains, and the code for collection and processing will be released in the supplementary materials. Additionally, the experiments in *Question 1* demonstrate that Fact-R1 has shown good applicability in the medical domain using the Covid-VTS [1] dataset.
>
> (3) Our long-term goal is to build and release a large-scale, fully open-source video dataset encompassing political, scientific, and healthcare information to support research on various types of misinformation manipulations. This effort aims to address the current gap in high-quality open-source datasets in the field. We are committed to continually updating and maintaining this dataset to foster the advancement of video misinformation detection.
>
>
> [1] Liu et al.. Covid-vts: Fact extraction and verification on short video platforms.
>
> [2] Park et al.. DeepVideo-R1: Video Reinforcement Fine-Tuning via Difficulty-aware Regressive GRPO.
>
> [3] Feng et al. Video-r1: Reinforcing video reasoning in mllms.
>
> [4] Yu et al. Dapo: An open-source llm reinforcement learning system at scale.
>
> [5] Liu et al. Ocrbench: on the hidden mystery of ocr in large multimodal models.
>
> [6] Guan et al. Hallusionbench: an advanced diagnostic suite for entangled language hallucination and visual illusion in large vision-language models.
>
> [7] Lu et al. Mathvista: Evaluating mathematical reasoning of foundation models in visual contexts.

---

> ### Comment · Reviewer_19bT · 2025-08-06
>
> Thank you for the detailed responses. it resolves most of my concerns.
> The additional evaluations in the medical domain and multiple benchmarks support the generalization claim, and the computational breakdown addresses efficiency and reproducibility concerns. The response is convincing, and clarifications on dataset expansion resolve my earlier concerns. I will keep my final score.

---

> > ### Author Response · Authors · 2025-08-06
> >
> > We sincerely appreciate your thoughtful follow-up and the time you dedicated to revisiting our response. Your engagement with our work is truly valued and deeply appreciated.

---

### Official Review · Reviewer_7T2b · 2025-07-02

**Clarity:** 3
**Significance:** 4
**Originality:** 3
**Rating:** 5
**Confidence:** 4

**Summary:**

This paper studies the misinformation detection in the videos. This task is challenging due to two facts: 1. There is no large scale training datasets for support the chain-of-thought reasoning for misinformation detection. 2. There is no chain-of-thought MLLM that can excel this misinformation detection task. To address those two challenges, the paper first collects a large-scale datasets for the misinformation detection. The detailed caption and the audio is sent to the DeepSeek-R1 to generate the chain-of-thought thinking and the answer. The detailed caption is generated by using GPT-4o. To train the misinformation detection chain-of-thought model, the paper first leverages DPO to train a policy model with human preference alignment. Then GRPO with specifically designed tasks (e.g. News Video Caption and News Image OCR) are leveraged to further train the policy model. The proposed model acheives higher performance than the baseline approach on three benchmarks.

**Questions:**

I have several questions about the dataset construction and the ablation studies.

1. For dataset construction: L187-188, what are the prompts used for Deepseek-R1?
2. L191: It is easy to do answer verification. I wonder how to conduct the chain-of-thought verification specifically? How to check the reasoning trace is correct?
3. For DPO, I understand this is to align with human preference. But I wonder whether this step is truly necessary? Given the ablation studies (Table 3), it seems w/o DPO, the model achieves quite good performance.
4. For Table2 domain specific baseline. It is great to see BERT and VIT are leveraged as modality specific baseline. However, BERT is not an instruction following LLM. It merely fills the missing components in sentences. I wonder how to conduct the baseline studies by using BERT explicitly?
5. Similar question above, I might miss some details, I wonder how to conduct the baseline studies with ViT? ViT is just a visual encoder, I wonder how would the Acc and F1 being calculated for the ViT model?
6. One missing piece is the dataset scale studies. I wonder whether the current dataset size is enough or we still need more datasets to address the misinformation detection task. Specifically, I wonder whether the size of the dataset is enough and whether the topic coverage of the dataset is sufficient.

**Ethical Concerns:**

["NO or VERY MINOR ethics concerns only"]

**Limitations:**

yes

**Quality:**

3

**Strengths And Weaknesses:**

Strength:
1. The large scale dataset is one of the major contributions. In the multimodal LLM domain, it is critical to have a large scale and high quality dataset for training and finetuing model. It is glad to see a carefully design and curated dataset.
2. The ablation studies are conducted carefully with the modality speicfic baselines. Although I have some questions for how those modality specific baselines are being conducted, overall it is a quite careful design.
3. The proposed approach achieves better performance than the baselines.

Weakenss:
1. Lacking some dataset scale studies.
2. It seems not very clear whether chain-of-thought is strictly required for this task, although the model trained with GRPO achieves better perfomrance based on the ablation studies.

---

> ### Author Rebuttal · Authors · 2025-07-31
>
> Thank you very much for your careful reading of the manuscript and your constructive comments, which have been extremely helpful in improving our work. Below, we provide point‑by‑point responses to your feedback.
>
> >**Q1:** What are the prompts used for Deepseek-R1?.
>
> **A:** We apologize for the omission. The exact prompts for DeepSeek‑R1 are provided in the supplementary material (Section D, L647–L651), and Figure 27 further illustrates the prompt used to guide the model in determining the real/fake label. We will explicitly reference these prompts in the main text in the revised manuscript for clarity.
>
> >**Q2:** I wonder how to conduct the chain-of-thought verification specifically? How to check the reasoning trace is correct?
>
> **A:** We apologize for the lack of detail in the original manuscript. Below we clarify how we verify CoT reasoning.
> After Step 1 (label filtering), some Long CoT instruction tuning samples may still have correct labels but incorrect reasoning (e.g., Figure 21, where the model predicts the right label but fails to identify the manipulated entity).
> To address this, we add an explainability filtering step. Using the prompt in Figure 25, GPT-4o-mini acts as a simulated human judge to check whether the reasoning correctly identifies the manipulation. As shown in Figures 17–18, its evaluations align closely with human judgments.
> This two stage filtering yields a high quality set of 85k Long CoT samples, ensuring that the final training data includes both correct labels and faithful reasoning traces.
>
> >**Q3:** For DPO, I understand this is to align with human preference. But I wonder whether this step is truly necessary?
>
> **A:** (1) Thank you for your question. The original Fact-R1 pipeline did not include a DPO phase. Our initial SFT + GRPO approach struggled to produce reasoning traces aligned with human reviewer logic and was prone to errors. To address this, we curated 5k human‑annotated preference samples targeting common failure cases—such as incorrect answers, hallucinated entities, and commonsense inconsistencies. These samples simulate the decision‑making process of human auditors, guiding the model to produce more realistic and human‑aligned reasoning.
>
> |Dataset|Method|Score|
> |:-:|:-:|:-:|
> |Interpretability|w/o DPO|71.9|
> ||Fact‑R1|76.4|
> |HallusionBench|w/o DPO|44.7|
> ||Fact‑R1|48.8|
>
> (2) As shown in the Table, DPO also improves interpretability accuracy and boosts performance on HallusionBench [1], which evaluates hallucination. Importantly, DPO uses human edited model outputs, reducing annotation workload and mitigating issues from inconsistent annotation styles.
>
> Overall, the DPO phase is crucial for validating and enhancing chain of thought reasoning, enabling Fact-R1 to produce more accurate and reviewer aligned explanations.
>
> >**Q4:** I wonder how to conduct the baseline studies by using BERT explicitly?.
>
> **A:** (1) We sincerely apologize for the lack of detail regarding the BERT baseline training in the main text. In our setup, we use BERT to extract textual features from news titles. Specifically, we take the [CLS] token embedding and feed it into an MLP classifier for binary classification, truncating input sequences to a maximum length of 512 tokens. BERT and the MLP are trained jointly using the same training data as the baseline methods, which include the training splits of FakeSV, FakeTT, and FakeVV. A classification threshold of 0.5 is applied to compute Accuracy and F1 scores.
>
> (2) As shown in Table 2 of the manuscript, BERT performs reasonably well on the real world datasets FakeSV and FakeTT, where misinformation often appears through anomalous tone or unusual symbols in the titles. However, its performance degrades significantly on FakeVV, highlighting the difficulty of detecting fine grained textual manipulations and the necessity of incorporating multimodal reasoning.
> These training details will be added to the final version of the manuscript for completeness.
>
>
> >**Q5:** ViT is just a visual encoder, I wonder how would the Acc and F1 being calculated for the ViT model?
>
> **A:** We sincerely apologize for not providing sufficient details on the ViT baseline training in the main text.
>
> (1) For keyframes, we used the ViT visual encoder to extract visual features from multiple frames. We then extracted the [CLS] token vector from each frame, averaged these vectors, and concatenated the averaged visual features with the [CLS] token vector from the pre-trained BERT. These concatenated features were then input into an MLP classifier.
> During training, we jointly trained ViT and MLP, while keeping BERT frozen. The training data and setup were consistent with the baseline, using the training sets from FakeSV, FakeTT, and FakeVV, with a 0.5 threshold for binary classification to compute Accuracy and F1 scores.
> As shown in Table 2 of the manuscript, the ViT baseline performed poorly across all three datasets, indicating that relying solely on misaligned visual features is insufficient for video misinformation detection and can even degrade performance.
>
> (2) To further validate the importance of visual-text alignment, we added a CLIP baseline experiment. We used pre-trained and frozen CLIP vision and text encoders to extract the averaged visual keyframe features and news title features. The features were concatenated and then fed into a trained MLP classifier. The training data and threshold settings remained consistent.
>
> |Dataset|Method|Acc|F1|
> |:-:|:-:|:-:|:-:|
> |FakeSV|CLIP|68.7|66.9|
> |FakeTT|CLIP|69.2|68.8|
> |FakeVV|CLIP|70.0|70.1|
>
> As shown in the results, the experiment confirmed the importance of introducing pre-trained visual-text alignment models.
> We apologize again for the earlier omission, and we will include these progressive baseline design details and findings in the revised manuscript.
>
>
> >**Q6:** One missing piece is the dataset scale studies.
>
> **A:** We thank the reviewer for this valuable suggestion, which prompted us to further reflect on improving the FakeVV dataset. We plan to expand FakeVV from two perspectives: source diversity and content diversity.
>
> (1) **Scaling dataset size and source diversity.** The current FakeVV dataset contains over 100k high-quality video–text pairs from four official news channels, creating a resource similar to VisualNews [2] in the video domain, with the goal of advancing the community's technical development. We plan to expand FakeVV by adding more outlets (e.g., Fox News, ABC News) to increase source diversity. Our analysis shows health‑ and science‑related topics account for only 13.6% and 7.9% of samples, while most focus on social, political, and cultural issues. To address this imbalance, we will add more video–text pairs from these underrepresented categories. FakeVV will remain fully open‑source, and we will continue to update and maintain it to support large‑scale video misinformation research.
>
> (2) **Expanding manipulation types and topic coverage.** Our novel non-random entity-replacement pipeline is highly extensible to other types of misinformation. Using the retrieval-based pipeline described in Figures 8–11, we have already constructed benchmarks for (a) temporal manipulation and (b) context manipulation without entity changes (*see the experiments in Question 3 to Reviewer B2W7*). The results show that Fact-R1 generalizes well to these cases, demonstrating its strong generalization ability in handling various manipulation types.
>
> (3) Moreover, we further enriched FakeVV with new manipulations, such as (c) numeric changes in news titles. For instance, the title  “Woman has over 1000 butt surgeries”  was modified to  “Woman has over 10000 butt surgeries” . If no retrieved titles contained numeric information, we performed random numeric substitutions. This resulted in 500 manipulated samples paired with 500 real cases.
>
> |Dataset|Method|Acc|F1|
> |:-:|:-:|:-:|:-:|
> |Number|GPT-4o|55.3|53.2|
> ||FakingRec|61.4|60.6|
> ||Fact-R1|67.7|65.5|
>
>
> As shown in the Table，Experimental results reveal that such manipulations remain challenging, motivating us to enhance Fact-R1’s ability to reason about numeric information.
> All enriched benchmarks and analyses will be released publicly to further support open source model training and research.
>
> >**Q7:** It seems not very clear whether chain-of-thought is strictly required for this task.
>
> **A:** We sincerely thank the reviewer for raising this important question.
>
> (1) **Qualitative Motivation for Chain‑of‑Thought Reasoning.** In real‑world misinformation auditing, reviewers require not just binary labels but transparent reasoning to understand and verify model outputs. Compared to many tasks in the community, misinformation detection relies more heavily on a verifiable and explainable decision-making process. Therefore, incorporating strong reasoning capabilities into open-source models is especially important, as it helps enhance the model's value in practical applications.
>
> (2) **Quantitative Benefits of Chain-of-Thought Reasoning.** In addition to significantly improving prediction performance, the ablation study in the table below demonstrates that the introduction of GRPO-based chain-of-thought reasoning training leads to a substantial improvement in interpretability accuracy.
>
> |Dataset|Method|Acc|
> |:-:|:-:|:-:|
> |Interpretability|w/o GRPO|67.1|
> ||Fact-R1|76.4|
>
> This shows that Fact‑R1 demonstrates emergent reasoning abilities similar to advanced text‑based RL systems, but in the more challenging context of video misinformation detection. Overall, the results highlight that chain‑of‑thought reasoning is particularly valuable for misinformation detection, where transparent and verifiable reasoning is essential for real‑world use.
>
>
> [1] Guan et al. Hallusionbench: an advanced diagnostic suite for entangled language hallucination and visual illusion in large vision-language models
>
> [2] Liu et al. 2021. Visual News: Benchmark and Challenges in News Image Captioning.

---

### Official Review · Reviewer_B2W7 · 2025-07-03

**Clarity:** 3
**Significance:** 2
**Originality:** 2
**Rating:** 3
**Confidence:** 4

**Summary:**

This paper introduces Fact-R1, a new reasoning model for detecting video misinformation, and an accompanying dataset, FakeVV. The FakeVV dataset is a large-scale benchmark containing over 100,000 video-text pairs, created by sourcing videos from news channels and synthetically generating misinformation by replacing entities (e.g., persons, locations) in the titles to create semantic inconsistencies with the video content. The proposed model, Fact-R1, is trained using a three-stage pipeline: (1) Supervised fine-tuning (SFT) on long-chain-of-thought (CoT) instructions, (2) Direct Preference Optimization (DPO) for preference alignment, and (3) Group Relative Policy Optimization (GRPO) with a custom rule-based reward function to enhance reasoning. The authors position this work as a new paradigm for explainable misinformation detection by combining video understanding with deep reasoning.

**Questions:**

1. The core training paradigm (SFT, DPO, GRPO for reasoning) is heavily inspired by DeepSeek-R1. Could the authors clarify the key methodological novelty of Fact-R1 beyond the application of this existing pipeline to the video-text modality?
2. The term "misinformation detection" implies an ability to check facts against reality. Given that Fact-R1 operates on a static dataset with no connection to external knowledge sources, how do the authors justify this claim? How would the model handle a news video about an event that occurred in March 2025? Wouldn't it be more accurate to frame the task as "cross-modal consistency checking"?
3. Generalization Beyond Entity Replacement: Fact-R1 is trained specifically on entity replacement misinformation (person, location, event, organization). How does the model perform on other common video misinformation types such as: (a) Temporal manipulation (old footage presented as recent), (b) Context manipulation without entity changes, (c) AI related misinformation like Deepfakes or other synthetic content, (d) Misleading captions on authentic footage? Could you provide experiments or analysis showing Fact-R1's performance on these scenarios?
4. Baseline Comparison Fairness and Real-world Performance: Several concerns about the experimental setup: (a) Many baseline models appear to be evaluated zero-shot or with suboptimal configurations - could you ensure fair comparison by providing task-specific fine-tuning for key baselines (at least for open-source models)? (b) While Fact-R1 does not currently support external search or retrieval, this is a critical limitation in real-world misinformation detection. There is no technical barrier preventing the model from calling a search API, and evaluating models that do so (e.g., retrieval-augmented systems) would offer a more complete understanding of Fact-R1’s practical utility.
5. Missing Strong Internet-Augmented Baseline (e.g., Grok-3 with internet search): The current baselines omit strong internet-connected models like Grok-3, which has been fine-tuned on misinformation discrimination tasks and has demonstrated strong performance on platform X.

**Ethical Concerns:**

["Major Concern: Improper research involving human subjects", "Major Concern: Discrimination, bias, and fairness", "Major Concern: Human rights (including surveillance)"]

**Final Justification:**

I thank the authors for their detailed and thorough rebuttal, which addressed most of my experimental setup concerns — including fairness in baseline comparisons, evaluation on additional manipulation types, and clarification of retrieval‑augmented baselines. These revisions substantially improve the completeness and transparency of the empirical evaluation.

However, my primary concern regarding limited methodological novelty remains. Fact‑R1’s three‑stage pipeline (SFT → DPO → GRPO) closely follows the DeepSeek‑R1 reasoning RL framework, with adaptations to a multimodal video–text setting. While these adaptations (task‑specific rewards, auxiliary tasks, dataset integration) are valuable engineering contributions, they do not amount to a fundamentally new learning paradigm. The main intellectual contribution here is the FakeVV dataset, which is large, well‑curated, and likely to be useful to the community — but dataset contribution alone does not, in my view, fully compensate for the lack of novel algorithmic ideas.

Given this balance, I raise my score to borderline reject, with the main reason being the lack of a sufficiently novel technical contribution despite strong dataset and experimental work.

**Limitations:**

The authors should expand the limitations section to address several critical concerns:

Dataset Scope: The FakeVV dataset only covers entity replacement misinformation from four Western news sources, which may not generalize to other manipulation types (deepfakes, temporal manipulation, context changes) or diverse cultural/linguistic contexts. The synthetic nature of the fake content may not reflect real-world misinformation patterns.
Reasoning Quality: The paper doesn't adequately address the risk of confident-sounding but factually incorrect explanations that could mislead users more than simple classification outputs, or how to distinguish genuine reasoning from sophisticated pattern matching.

**Quality:**

3

**Strengths And Weaknesses:**

Strengths
The primary strength of this work is the introduction of FakeVV, a large-scale, diverse, and well-annotated benchmark dataset for video misinformation detection. With over 100,000 video-text pairs and fine-grained manipulated-entity annotations, FakeVV significantly improves the resources available for this underexplored task. Its inclusion of interpretable labels and support for explainable evaluation makes it a valuable contribution to the community.

Weaknesses:
1. Limited Methodological Novelty: The core training methodology (SFT -> DPO -> GRPO) is a direct application of techniques developed for text-only Large Reasoning Models, most notably DeepSeek-R1, which the authors themselves cite as inspiration. While applying this to the multimodal video domain is a valid engineering contribution, the paper lacks a fundamental novel algorithm or framework. The contribution is more an application of an existing reasoning RL pipeline to a new domain rather than a new method for misinformation detection itself.
2. Static Knowledge and Misleading "Misinformation" Claim: A critical flaw is that the system operates in a closed world. Fact-R1 cannot verify information against the real, evolving world, as it has no connection to the internet or any live knowledge base. Its training data extends only to February 2025. This means the model is incapable of detecting misinformation about any event occurring after this date. The task it solves is not "misinformation detection" in a general sense, but rather "detecting semantic inconsistency between a video and its accompanying static title." This is a much narrower and less impactful problem. True misinformation often requires external fact-checking, which this model cannot perform.
3. "Explainability" is Not Verifiable: The paper claims its method is "explainable" because it generates a Chain-of-Thought (CoT) reasoning trace. However, this "explanation" is generated by the same model making the prediction and is prone to hallucination—producing a plausible-sounding justification for a potentially incorrect conclusion. There is no guarantee that the reasoning trace reflects the model's actual causal process. Evaluating these traces with another LLM (GPT-40-mini) is circular and does not constitute a robust verification of the explanation's faithfulness.

---

> ### Author Rebuttal · Authors · 2025-07-31
>
> Thank you very much for your thoughtful and constructive feedback. We sincerely appreciate your recognition of FakeVV, which greatly enriches resources for this underexplored task. Below are our point by point responses to your comments.
> >**Q1:** Limited Methodological Novelty...
>
> **A:** While our training paradigm is inspired by DeepSeek-R1, Fact-R1 is not a simple extension to video-text tasks but a task-specific framework designed for video misinformation detection:
>
> (1) **Research Motivation and Dataset Contribution:** In real world video auditing, users need not only labels but also verifiable reasoning traces. To address this, we introduce FakeVV for the first time, a dataset of 100k video–text pairs with fine grained metadata, providing a video domain resource to advance community research.
>
> (2) **Generalizable Data Pipeline:** Our non-random entity replacement pipeline can naturally extend to scenarios such as temporal and context manipulation, addressing the evolving challenges of video misinformation.
>
> (3) **Task-Specific Rewards & Auxiliary Tasks:** Fact-R1 introduces novel verifiable reward functions, including entity recognition ($R_{\text{entity}}$) and reasoning keywords ($R_{\text{word}}$), guided by a judge model to generate correct reasoning chains. Inspired by Figure 20, we also incorporate News Image OCR and video caption generation as auxiliary tasks with dedicated rewards. As shown in Table 4 of the manuscript, these domain specific designs substantially improve Fact-R1’s reasoning ability, making it a specialized framework rather than a direct reuse of existing RL pipelines.
>
> >**Q2:** The term "misinformation detection" implies an ability to check facts against reality…
>
> **A:** We sincerely thank the reviewer for their valuable feedback, which has prompted deeper reflection on the task definition and guided us in further improving Fact-R1 for video misinformation detection.
>
> (1) **Definition Consistency.** Our task definition follows prior work [1,2], which treats static multimodal datasets as benchmarks for misinformation detection. To ensure fairness and comparability, we strictly adopted the baseline setups of widely used datasets such as FakeSV  and FakeTT . Results on both FakeVV and these real world datasets show that Fact-R1 performs strongly in video misinformation detection.
>
> (2) **Generalization Beyond Training Data.** In fact, Fact-R1 was trained solely on samples dated before 2025 (**not including February 2025**), while the FakeVV test set contains events from 2025. As shown in Figure 19 (Case 1), the model accurately detects manipulations and produces valid reasoning for unseen events, indicating that it learns generalizable reasoning patterns rather than simply memorizing facts.
>
> (3) **Scalability to Fact‑Checking.** While trained on static data, Fact‑R1 is model‑agnostic and can integrate naturally with retrieval‑based fact‑checking pipelines. Using a retrieval augmentation method similar to 3MFact [4], retrieval significantly improved performance, as shown in the table below. This demonstrates that Fact‑R1 can serve as the reasoning core of a dynamic fact‑checking system enhanced by external evidence.
>
> |Dataset |Method|Acc|F1|
> |:-:|:-:|:-:|:-:|
> |FakeSV|Fact‑R1|75.6|74.7|
> ||Fact‑R1+Retrieval|79.5(↑3.9)|78.3(↑3.6)|
> |FakeTT|Fact‑R1|74.4|72.7|
> ||Fact‑R1+Retrieval|79.4(↑5.0)|78.9(↑6.2)|
> |FakeVV|Fact‑R1|81.2|80.3|
> ||Fact‑R1+Retrieval|86.5(↑5.3)|84.5(↑4.2)|
>
> >**Q3:** Generalization Beyond Entity Replacement.
>
> **A:** To assess generalization beyond entity replacement, we constructed four representative benchmarks:
> - (a) **Temporal Manipulation:** 500 samples from FakeVV with time references were modified by replacing timestamps using retrieved news titles (Figures 8–11). As shown in *Table Temporal*, this task is challenging, and Fact‑R1 performs well mainly when clear temporal cues are present.
> - (b) **Context Manipulation without Entity Changes:** For 500 real samples, titles were replaced with mismatched real titles (following the procedure described in L604–L624 of the manuscript). *Table Context* shows that Fact‑R1 detects these out‑of‑context inconsistencies effectively.
> - (c) **AI‑related Manipulation (Deepfake):** Using FaceSwap [3], we replaced the largest face with a random CelebA‑HQ face to generate 500 forged samples. *Table Deepfake* demonstrates Fact‑R1’s strong generalization to synthetic media.
> - (d) **Misleading captions on authentic footage:** We extracted 50 fake samples from the test set and used Photoshop to modify keyframe captions to match fake news titles, creating 50 enhanced fake samples. *Table Captions* shows a ~5% accuracy drop, indicating the higher difficulty of this scenario.
>
> |Dataset|Method|Acc|F1|
> |:-:|:-:|:-:|:-:|
> |Temporal|GPT‑4o|62.2|61.3|
> ||FakingRec|63.5|63.1|
> ||Fact‑R1|69.7|67.8|
> |Context|GPT‑4o|78.2|78.0|
> ||FakingRec|77.7|75.9|
> ||Fact‑R1|86.4|85.9|
> |Deepfake|GPT‑4o|64.5| 62.6|
> ||FakingRec|68.3|66.2|
> ||Fact‑R1|76.8|76.4|
> |Captions|GPT‑4o|68.0|61.9|
> ||FakingRec|64.0| 60.8|
> ||Fact‑R1|73.0(↓5.0) |71.7(↓4.4) |
>
> As noted in [1], “Unfortunately, most of the datasets are not released.” Inspired by this and the reviewer’s suggestion, we enriched FakeVV and proposed Fact‑R1 as a strong baseline with broad generalization.
>
> >**Q4:** Could you ensure fair comparison by providing task-specific fine-tuning for key baselines...
>
> **A:** We thank the reviewer for their valuable feedback on baseline fairness. In the revised version, we added fine-tuning experiments with open-source models, including InternVL2.5‑8B and Qwen2.5‑VL‑7B, using the same training data from FakeSV, FakeTT, and FakeVV with label-based answers. The experiments used the same LoRA parameters as Fact‑R1.
>
> |Dataset|Method|Acc|F1|
> |:-:|:-|:-:|:-:|
> |FakeSV|Qwen2.5-VL-7B-SFT|71.8|72.1|
> ||InternVL2.5-8B-SFT|70.1|66.1|
> ||Fact-R1|75.6|74.7|
> |FakeTT|Qwen2.5-VL-7B-SFT|68.8|66.4|
> ||InternVL2.5-8B-SFT|67.7|68.0|
> ||Fact-R1|74.4|72.7|
> |FakeVV|Qwen2.5-VL-7B-SFT|75.4|73.3|
> ||InternVL2.5-8B-SFT|73.5|73.2|
> ||Fact-R1|81.2|80.3|
>
> As shown in the Table, fine tuning Qwen2.5-VL-7B achieved strong performance, further confirming its effectiveness as the backbone for Fact-R1.
>
> >**Q5:** And evaluating models that do so (e.g., retrieval-augmented systems) ... Missing Strong Internet-Augmented Baseline..
>
> **A:**  Following the fact‑checking setup [5], we adopted the evidence retrieval process from 3MFact [4] to enhance Fact‑R1. As shown in the table, the retrieval‑enhanced Fact‑R1 outperforms other methods, confirming its potential as the reasoning core of fact‑checking systems.
> We also evaluated Grok‑3‑deepsearch and Grok‑4, which achieved strong zero‑shot performance, demonstrating the power of retrieval‑augmented models. However, unlike image fact‑checking, video fact‑checking incurs high API costs and raises privacy concerns. As noted in 3MFact [4], GPT‑4o‑mini was used as the LLM and tested on only a small number of samples, underscoring the need for efficient and accessible open‑source models.
>
> |Dataset|Method|Acc|F1|
> |:-:|:-|:-:|:-:|
> |FakeSV|Grok-3-deepsearch|68.8|69.2|
> ||Grok-4|72.4|72.6|
> ||3MFact|71.5|71.0|
> ||Fact-R1|75.6|74.7|
> ||Fact-R1+Retrieval|79.5(↑3.9)|78.3(↑3.6)|
> |FakeVV|Grok-3-deepsearch|69.2|70.1|
> ||Grok-4|75.5|74.8|
> ||3MFact|77.7|76.3|
> ||Fact-R1|81.2|80.3|
> ||Fact-R1+Retrieval|86.5(↑5.3)|84.5(↑4.2)|
>
>
> >**Q6:** "Explainability" is Not Verifiable...
>
> **A:** We thank the reviewer for this insightful question, which helps clarify our approach to explainability.
>
> (1) **Motivation.** In real-world misinformation auditing, reviewers need not only labels but also transparent reasoning. This aligns with Bu et al. [1], who emphasize predicting specific misinformation types with fine-grained annotations. FakeVV supports this by encouraging reasoning tied to manipulation types.
>
> (2) **Credibility.** Instead of style-sensitive metrics (e.g., ROUGE), FakeVV allows quantitative evaluation via entity-level accuracy. During GRPO, a judge model simulates human reviewers to guide reasoning. Table 4 in the manuscript shows that removing the judge lowers performance, and Figures 17–18 confirm GPT‑4o‑mini as judge aligns well with human evaluation.
>
> >**Q7:** Limitations
>
> **A:**(1) **Dataset and Generalization:** We provide an open-source data pipeline that generalizes to diverse manipulation types. Experiments show Fact‑R1 performs well on temporal, contextual, deepfake, and misleading captions.
>
> (2) **Real-World Performance:** Fact‑R1 achieves strong results on FakeSV and FakeTT. Retrieval augmentation further improves factual alignment, showing its reasoning goes beyond memorization.
>
> (3) **Incorrect Reasoning Risk:** As discussed in the original manuscript, current AI auditing models—like human auditors—are not flawless. While Fact‑R1 may occasionally produce confident but incorrect reasoning, these traces are valuable as diagnostic tools, helping human reviewers identify failure cases and guide model improvement. As shown in Figure 21, such errors are often easily detected by humans. We plan to integrate Fact‑R1 into a human‑AI collaborative auditing workflow, where its reasoning outputs assist reviewers, ensuring reliability in practical use.
>
> >**Q8:** Major Concern: Improper research involving human subjects... Major Concern: Human rights (including surveillance)
>
> **A:** We apologize for any concerns and confirm that this work involves no human-subject research, discrimination, bias,  fairness, human-rights issues, or surveillance.
>
> [1] Bu et al. Combating online misinformation videos: Characterization, detection, and future directions.
>
> [2] Shao et al. Detecting and grounding multi-modal media manipulation.
>
> [3] FaceSwap Contributors. Faceswap: Deepfakes software for face swapping. GitHub repository.
>
> [4] Niu et al. Explainable Video Fact-Checking with a New Dataset and Multi-role Multimodal Model Approach.
>
> [5] Braun et al. Defame: Dynamic evidence-based fact-checking with multimodal experts.

---

> > ### Author Response · Authors · 2025-08-01
> >
> > Dear Reviewer B2W7,
> >
> > We sincerely appreciate the time and thoughtful feedback you have provided. We have thoroughly addressed the Ethical Concerns raised by you, as well as by Ethics Reviewers dQpE, P7BD, and uGv2, and have improved the related discussions in the manuscript. Regardless of the final outcome, we greatly value your feedback and sincerely thank you for your engagement and support of our work.
> >
> > Sincerely,
> > The Authors

---

### Decision · Program_Chairs · 2025-09-17

**Decision:**

Accept (poster)

**Comment:**

The manuscript investigates detecting video misinformation, proposing a reasoning model and a large-scale benchmark comprising over 100k video-text pairs. The dataset is constructed by sourcing videos from from the web (e.g., news channels), whereas the proposed model leverages a three-stage pipeline.  The manuscript received ratings of three accept and one borderline reject. Reviewers appreciated the proposed dataset, extensive ablation studies, and performance of the proposed model against the baseline. Reviewers also raised several questions about dataset construction and ablation studies along with one reviewer raisning ethical concerns. While the ethics reviewers acknowledged some of these concerns, they generally consider these concerns to be addressable suggesting recommendations including, providing more details (Ethics Reviewer dQpE), addressing discrepancies in the checklist along with moving limitations (Appendix F) and societal impacts (Appendix G) to the main paper (Ethics Reviewer P7BD), and discussing potential dataset bias in more details (Ethics Reviewer uGv2). Given that three reviewers are positive about the manuscript, authors response raised in the initial review, and overall reviews, the recommendation is accept. Authors are strongly encouraged to take into account all the recommendations of the reviewers in the final manuscript.